# YEATS2 links histone acetylation to tumorigenesis of non-small cell lung cancer

Wenyi Mi [1,2], Haipeng Guan[3,4], Jie Lyu [5], Dan Zhao[3,4], Yuanxin Xi[5], Shiming Jiang[1,2], Forest H. Andrews[6], Xiaolu Wang[1,2], Mihai Gagea [7], Hong Wen[1,2], Laszlo Tora [8,9,10,11], Sharon Y.R. Dent[1,2,12], Tatiana G. Kutateladze[6], Wei Li [5], Haitao Li [3,4] & Xiaobing Shi [1,2,12]

Recognition of modified histones by "reader" proteins constitutes a key mechanism regulating diverse chromatin-associated processes important for normal and neoplastic development. We recently identified the YEATS domain as a novel acetyllysine-binding module; however, the functional importance of YEATS domain-containing proteins in human cancer remains largely unknown. Here, we show that the *YEATS2* gene is highly amplified in human non-small cell lung cancer (NSCLC) and is required for cancer cell growth and survival. YEATS2 binds to acetylated histone H3 via its YEATS domain. The YEATS2-containing ATAC complex co-localizes with H3K27 acetylation (H3K27ac) on the promoters of actively transcribed genes. Depletion of YEATS2 or disruption of the interaction between its YEATS domain and acetylated histones reduces the ATAC complex-dependent promoter H3K9ac levels and deactivates the expression of essential genes. Taken together, our study identifies YEATS2 as a histone H3K27ac reader that regulates a transcriptional program essential for NSCLC tumorigenesis.

[1] Department of Epigenetics and Molecular Carcinogenesis, The University of Texas M.D. Anderson Cancer Center, Houston, TX 77030, USA. [2] Center for Cancer Epigenetics, The University of Texas M.D. Anderson Cancer Center, Houston, TX 77030, USA. [3] MOE Key Laboratory of Protein Sciences, Beijing Advanced Innovation Center for Structural Biology, Department of Basic Medical Sciences, School of Medicine, Tsinghua University, Beijing 100084, China. [4] Tsinghua-Peking Joint Center for Life Sciences, Tsinghua University, Beijing 100084, China. [5] Department of Molecular and Cellular Biology, Dan L. Duncan Cancer Center, Baylor College of Medicine, Houston, TX 77030, USA. [6] Department of Pharmacology, University of Colorado School of Medicine, Aurora, CO 80045, USA. [7] Department of Veterinary Medicine & Surgery, The University of Texas M.D. Anderson Cancer Center, Houston, TX 77030, USA. [8] Institut de Génétique et de Biologie Moléculaire et Cellulaire, 67404 Illkirch, France. [9] Centre National de la Recherche Scientifique, UMR7104, 67404 Illkirch, France. [10] Institut National de la Santé et de la Recherche Médicale, U964, 67404 Illkirch, France. [11] Université de Strasbourg, 67404 Illkirch, France. [12] Genes and Development and Epigenetics & Molecular Carcinogenesis Graduate Programs, The University of Texas Graduate School of Biomedical Sciences, Houston, TX 77030, USA. Wenyi Mi, Haipeng Guan, Jie Lyu and Dan Zhao contributed equally to this work. Correspondence and requests for materials should be addressed to W.L. (email: wl1@bcm.edu) or to H.L. (email: lht@tsinghua.edu.cn) or to X.S. (email: xbshi@mdanderson.org)

Lysine acetylation is one of the most frequent post-translational modifications occurring on histones that play a critical role in regulating chromatin dynamics and the accessibility of the underlying DNA in eukaryotes[1]. Acetylation on histone lysine residues is controlled by two families of counteracting enzymes: histone acetyltransferases (HATs) and histone deacetylases (HDACs), and is normally associated with active transcription[2, 3]. In addition to neutralizing the positive charge on the side chain of lysine residues, the bulky acetyl groups can also serve as docking sites for reader proteins, which recognize this specific modification and transduce the molecular signals to elicit various downstream biological outcomes[4]. Bromodomain (BRD) has long been thought to be the sole protein module that specifically recognizes acetyllysine motifs[5]. Some tandem plant homeodomain zinc fingers were later found to bind histone H3 in an acetylation-sensitive manner[6–8]. Recently, we identified the YEATS domain of AF9 protein as a novel reader of histone acetylation[9]. YEATS domain is evolutionarily conserved from yeast to human[10]. There are four YEATS domain-containing proteins in humans and three in *Saccharomyces cerevisiae*[11]. All the YEATS domain proteins are associated with chromatin-associated complexes, such as HAT complexes and chromatin-remodeling complexes, however, the functions of these proteins, and particularly their YEATS domains, are not well understood.

YEATS domain-containing 2 (YEATS2) is a scaffolding subunit of the Ada-two-A-containing (ATAC) complex, a conserved metazoan HAT complex[12, 13]. Vertebrate ATAC complexes share the same catalytic HAT subunit, GCN5, or the highly related PCAF in mammals, with another multi-subunit complex Spt–Ada–Gcn5–acetyltransferase (SAGA)[14, 15]. Although the SAGA complex has been extensively studied in both yeast and humans, much less is known about the ATAC complex. GCN5 and PCAF in the ATAC complex mainly acetylate histone H3K9 and H3K14, while the second acetyltransferase ATAC2 in the complex has been reported to modify H4K16[16, 17]. The ATAC complex occupies distinct set of genes from SAGA and coordinates MAP kinases to regulate JNK target genes[18, 19]. The subunits of SAGA form four sub-modules that exert distinct molecular functions within the complex[20, 21], however, within the ATAC complex, except the HAT module, the functions of most other subunits remain largely unknown. In this study, we characterized the molecular and biological functions of YEATS2 within the ATAC complex. We found that the *YEATS2* gene is highly amplified in human cancers including non-small cell lung cancer (NSCLC). Depletion of YEATS2-reduced cancer cell growth, survival and transformation activity. The YEATS domain of YEATS2 binds to acetylated histone H3K27 (H3K27ac). Recognition of histone acetylation is important for the functions of YEATS2 in cells. Disruption of acetylation recognition of YEATS2-abrogated GCN5/PCAF-mediated promoter histone acetylation and consequently, suppressed the expression of its target genes, including the ribosomal protein-encoding genes that are essential for cell growth and survival. Taken together, our results identified YEATS2 as a histone H3K27ac reader that epigenetically regulates a transcriptional program essential for NSCLC tumorigenesis.

## Results

*YEATS2 is an essential gene amplified in NSCLC.* To determine whether YEATS2 plays a role in human cancers, we first examined *YEATS2* gene expression status across cancers in The Cancer Genome Atlas database via The cBioPortal for Cancer Genomics. As part of the 3q26 amplicon (Supplementary Fig. 1a), *YEATS2* is highly amplified in a variety of human cancers, including lung squamous cell carcinoma (56% amplification frequency),

ovarian serous cystadenocarcinoma (27%), and head and neck squamous cell carcinoma (23%) (Fig. 1a). Importantly, *YEATS2* gene expression levels are positively correlated to its amplification status in these tumors (Supplementary Fig. 1b–d). In human NSCLC and ovarian cancer patients, high *YEATS2* expression levels are correlated with worse prognosis (Supplementary Fig. 1e, f).

We next assessed YEATS2 expression levels across a number of lung cancer cell lines. Compared to the immortalized "normal" lung fibroblast cell lines (WI-38 and IMR-90), YEATS2 was overexpressed at both transcript and protein levels in all NSCLC cell lines we examined (Fig. 1b and Supplementary Fig. 2a). YEATS2 is a stoichiometric component of the ATAC HAT complex, which catalyzes histone acetylation, mainly on H3K9 and H3K14, by the enzymatic subunit GCN5 or PACF[12, 13]. Interestingly, compared with the immortalized normal cells, we also observed elevated levels of GCN5 and PCAF in most examined lung cancer cells (Fig. 1b), suggesting that essential subunits of the ATAC complex cooperate in human cancers likely leading to an super-active complex. Consistent with this speculation, we found global histone acetylation levels, especially H3K9ac, were evidently higher in the NSCLC cell lines than the immortalized normal cells (Fig. 1b). Interestingly, we also observed increased HDAC1 protein levels in cancer cells, which is opposite to the increased K9 acetylation (Fig. 1b).

Even though cancer cells acquire multiple genetic and epigenetic abnormalities, their growth and survival are often impaired by inactivation of a single oncogene. Since *YEATS2* is highly amplified in NSCLC, we sought to determine whether depletion of YEATS2 affects lung cancer cell growth. To this end, we knocked down (KD) *YEATS2* gene expression in the H1299 lung adenocarcinoma cell line using two independent shRNAs (Supplementary Fig. 2b) and determined cell growth. We observed a marked suppression of cell proliferation in cells treated with YEATS2-targeting shRNAs (shY2) compared with the cells treated with a non-targeting control shRNA (shNT) (Fig. 1c). Notably, the levels of suppression were correlated with the KD efficiency, with severe growth defect observed in the cells with higher KD efficiency. The growth inhibition by YEATS2 KD was also observed in additional NSCLC cell lines (A549, H520, and Ludlu-1) and ovarian cancer cell lines (CaoV3 and HeyA8) that also harbor *YEATS2* amplification, as well as in the immortalized normal lung fibroblast cells (WI-38 and IMR-90) that do not have *YEATS2* overexpression (Supplementary Fig. 2c–i), suggesting that *YEATS2* is an essential gene for a broad range of cancer cell lines as well as non-cancerous cells.

Cancer cells evolve with capability to undergo unlimited cell division and transformation. We next sought to test whether YEATS2 is required for cell survival and transformation of NSCLC. In clonogenic assay of both H1299 and A549 cells, the YEATS2 KD cells developed fewer colonies compared with the control cells, suggesting that YEATS2 is required for lung cancer cell survival (Fig. 1d and Supplementary Fig. 2j). We also performed soft agar colony formation assays to determine the effect of YEATS2 KD on anchorage-independent growth, an ability of transformed cells to grow independently of a solid surface[22]. Compared with the shNT treated control cells, YEATS2 KD resulted in fewer and also smaller colonies in soft agar in both H1299 and A549 cells (Fig. 1e and Supplementary Fig. 2k). Importantly, the defects associated with YEATS2 KD in both clonogenic and anchorage-independent cell growths were rescued by ectopic expression of shRNA-resistant YEATS2. Taken together, these results indicate that YEATS2 is required for cell growth, survival, and transformation of lung cancer cells.

**YEATS2 controls the expression of ribosome protein genes.** To determine how YEATS2 regulates cancer cell growth and survival, we performed RNA-seq analysis in YEATS2 KD cells to identify the genes regulated by YEATS2 genome-wide. We used YEATS2-targeting shRNA shY2-1 since this shRNA exhibited an efficient KD (Fig. 1c and Supplementary Fig. 2b), and we performed RNA-seq experiments in duplicates. We identified 1748 genes that were downregulated (false discovery rate (FDR) < 0.01), whereas 3361 genes upregulated, in YEATS2 KD cells compared with the control cells (Fig. 2a and Supplementary Data 1, 2). Kyoto

Encyclopedia of Genes and Genomes pathway analysis of the differentially expressed genes using DAVID (Database for Annotation, Visualization, and Integrated Discovery) revealed that the dysregulated genes were involved in vital biological processes, with downregulated genes enriched in the pathways regulating ribosome biogenesis, DNA replication, cell cycle, DNA repair, and splicing, whereas upregulated genes enriched in the pathways of lysosome functions, glycan degradation, and focal adhesion, etc (Fig. 2b and Supplementary Data 3). RNA-seq analysis using an independent shRNA (shY2-2) that partially KD

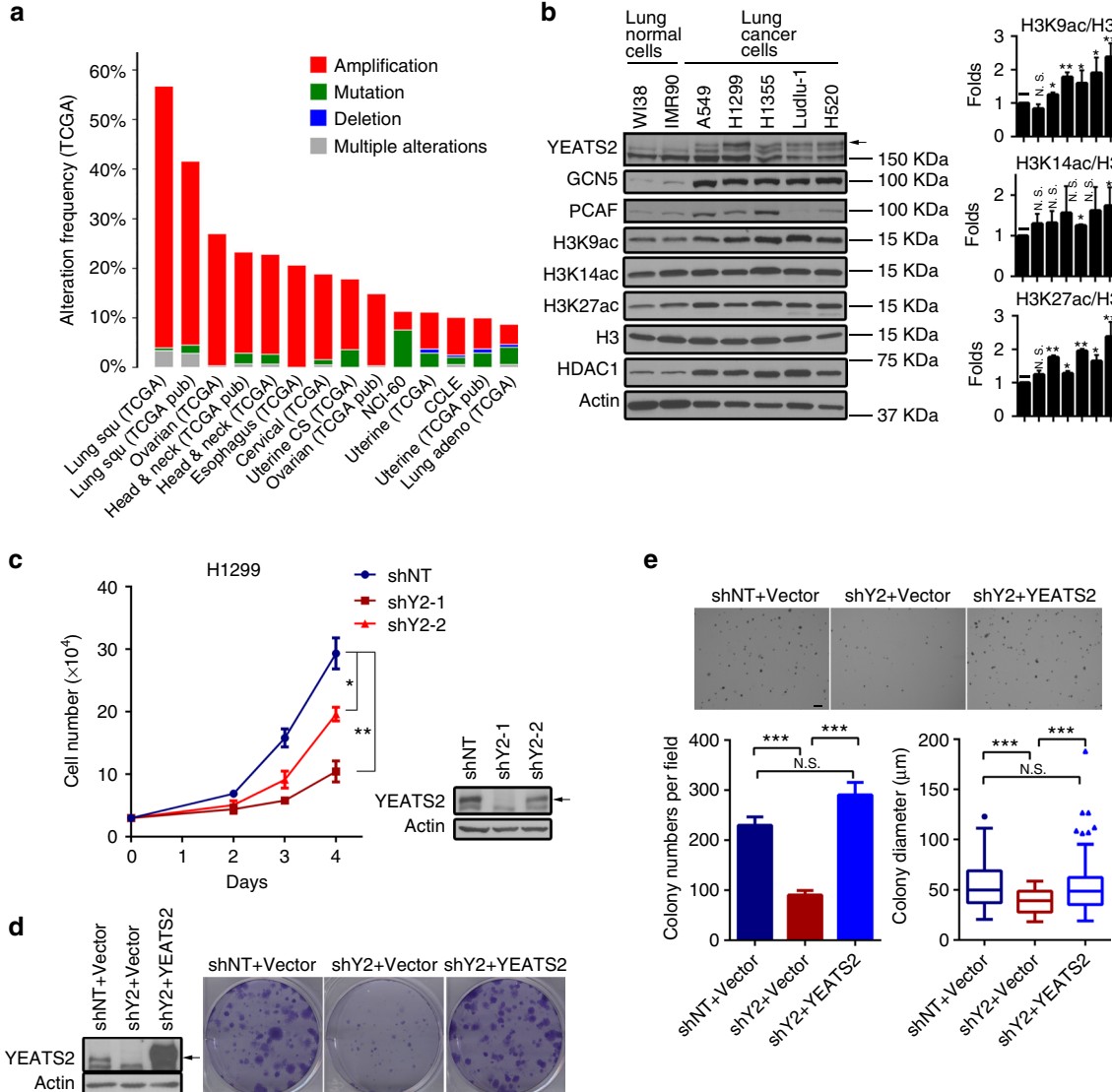

**Fig. 1** *YEATS2* is amplified in NSCLC and is required for cancer cell growth and survival. **a** *YEATS2* gene is frequently amplified in various types of human cancers. Data was obtained from the cBioPortal for Cancer Genomics. **b** Western blot analysis of YEATS2, GCN5, PCAF, HDAC1, and the indicated histone acetylation in NSCLC cell lines and immortalized "normal" lung fibroblast cell lines. Total H3 and actin are shown as loading control. The arrow indicates the band of YEATS2 protein. Relative H3K9ac, H3K14ac, and H3K27ac levels were quantified ($n = 3$, mean ± s.e.m.). N.S. not significant; *$p < 0.05$; **$p < 0.01$ (Student's *t*-test). **c** Cell proliferation assay of H1299 cells treated with control (shNT) or YEATS2 shRNAs (shY2). Cells (mean ± S.E.M., $n = 4$) were counted for 4 days after seeding (left panel). Right panel: western blot analysis showing YEATS2 knockdown efficiency. The arrow indicates the band of YEATS2 protein. Error bars represent S.E.M. *$p < 0.05$; **$p < 0.01$ (Student's *t*-test). **d** Clonogenic assay of control (shNT), YEATS2 knockdown (shY2), and knockdown H1299 cells rescued with ectopic expression of YEATS2. Empty vector was used as a control. Colonies were stained and photographed 7 days after seeding (right panel). Left panel: western blot analysis of YEATS2 expression level in indicated cells. The arrow indicates the band of YEATS2 protein. **e** Anchorage-independent growth assay of H1299 cells as in (**d**). Cells (mean ± S.E.M., $n = 4$-$6$) were stained with 0.005% crystal violet blue and photographed 3 weeks after seeding (top panel). Colony numbers (bottom left) and colony diameters (bottom right) were measured and quantified using ImageJ software. Scale bar, 200 μm. N.S. not significant; ***$p < 0.001$ (Student's *t*-test)

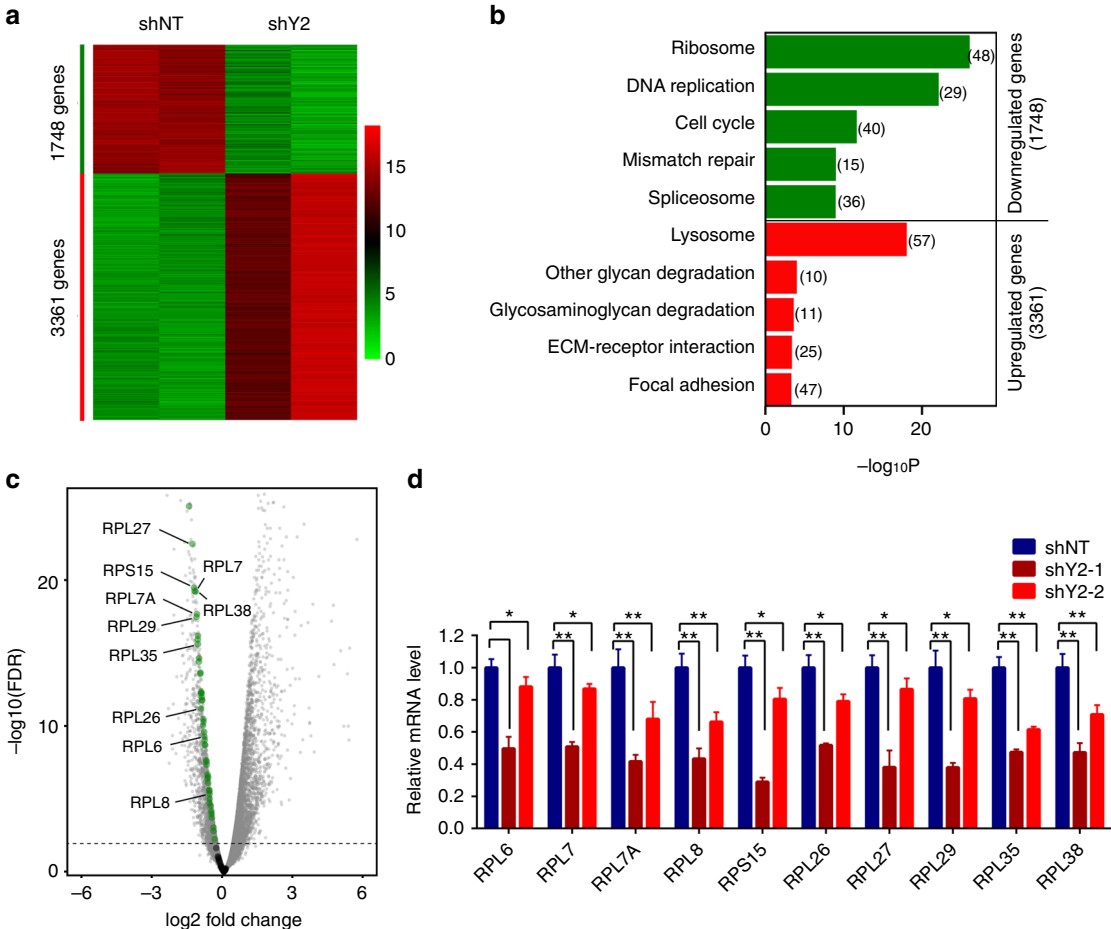

**Fig. 2** YEATS2 is required for the expression of ribosome protein-encoding genes. **a** Heatmap representation of differentially expressed genes in control (shNT) and YEATS2 knockdown (shY2) cells from two independent biological replicates of RNA-seq experiments. Fisher's exact test was used to define differentially expressed genes ($q < 0.01$). The color key represents normalized Log2 expression values. **b** Kyoto Encyclopedia of Genes and Genomes pathway analysis of downregulated (green) or upregulated (red) genes in YEATS2 knockdown cells compared with control cells. The numbers of genes within each functional group are shown in parenthesis. Fisher's exact test was used to identify the biological function with significant $p$-values (Benjamini–Hochberg corrected $p < 0.05$). **c** Volcano plot of differentially expressed genes in YEATS2 knockdown cells compared with control cells. 49 downregulated ribosomal protein genes are shown in green and 30 non-differentially expressed ribosomal protein genes in black. FDR, false discovery rate. **d** Quantitative RT-PCR (qRT-PCR) analysis of the expression of ten randomly picked ribosomal protein genes in control (shNT) and YEATS2 knockdown (shY2) cells. Error bars indicate S.E.M. of three biological replicates. *$p < 0.05$; **$p < 0.01$ (Student's $t$-test)

YEATS2 identified 1620 genes downregulated, among which significant number of genes (520), including 11 ribosomal protein genes, overlapped in both KD cells (Supplementary Fig. 3a, b). Notably, both downregulated and upregulated genes were enriched in pathways in cancers, including lung, colorectal, and pancreatic cancers (Supplementary Data 3), suggesting that YEATS2 controls growth and survival of various types of tumors through transcriptional regulation of essential genes.

Because YEATS2 is a subunit of the ATAC complex that is mostly involved in gene activation, we first focused on the genes downregulated by YEATS2 depletion in this study. The ribosome is a cellular machine for protein synthesis that is essential for sustained growth of both normal and cancer cells. Strikingly, among the genes encoding all 79 known ribosomal proteins, 49 genes were downregulated whereas none were upregulated in cells treated with YEATS2 shRNA (shY2-1) (Fig. 2c and Supplementary Data 1 and 2). Downregulation of these ribosomal protein-encoding genes was validated by qRT-PCR in cells treated with two YEATS2-targeting shRNAs, with levels of suppression correlated with KD efficiency (Fig. 2d). Flow cytometry analyses revealed that KD of YEATS2 led to G1 arrest of cell cycle, whereas

little or no defect in cell apoptosis or migration (Supplementary Fig. 3c–e), suggesting that growth suppression by YEATS2 KD is, at least in part, due to perturbation of cell cycle progression and DNA replication. Taken together, these results suggest that YEATS2 regulates the expression of genes involved in critical pathways such as ribosome biogenesis that are essential for maintaining cell growth and survival.

**The YEATS domain of YEATS2 binds to acetylated H3K27.** The recognition of histone H3 acetylation is an evolutionarily conserved function of the AF9 YEATS domain[9], we thus reasoned that the YEATS domain of human YEATS2 (Fig. 3a) may also binds to acetylated histones. To test this hypothesis, we performed histone peptide pulldown assays and we found that the YEATS domain of YEATS2 bound specifically to histone H3K27ac, with weak or no bindings to other acetylated histone peptides (Fig. 3b). Histone-binding assay in vitro and protein-chromatin binding in cells demonstrated the interaction between YEATS2 and H3K27ac at full-length histone and nucleosomal levels, respectively (Fig. 3c, d). Quantitative isothermal titration

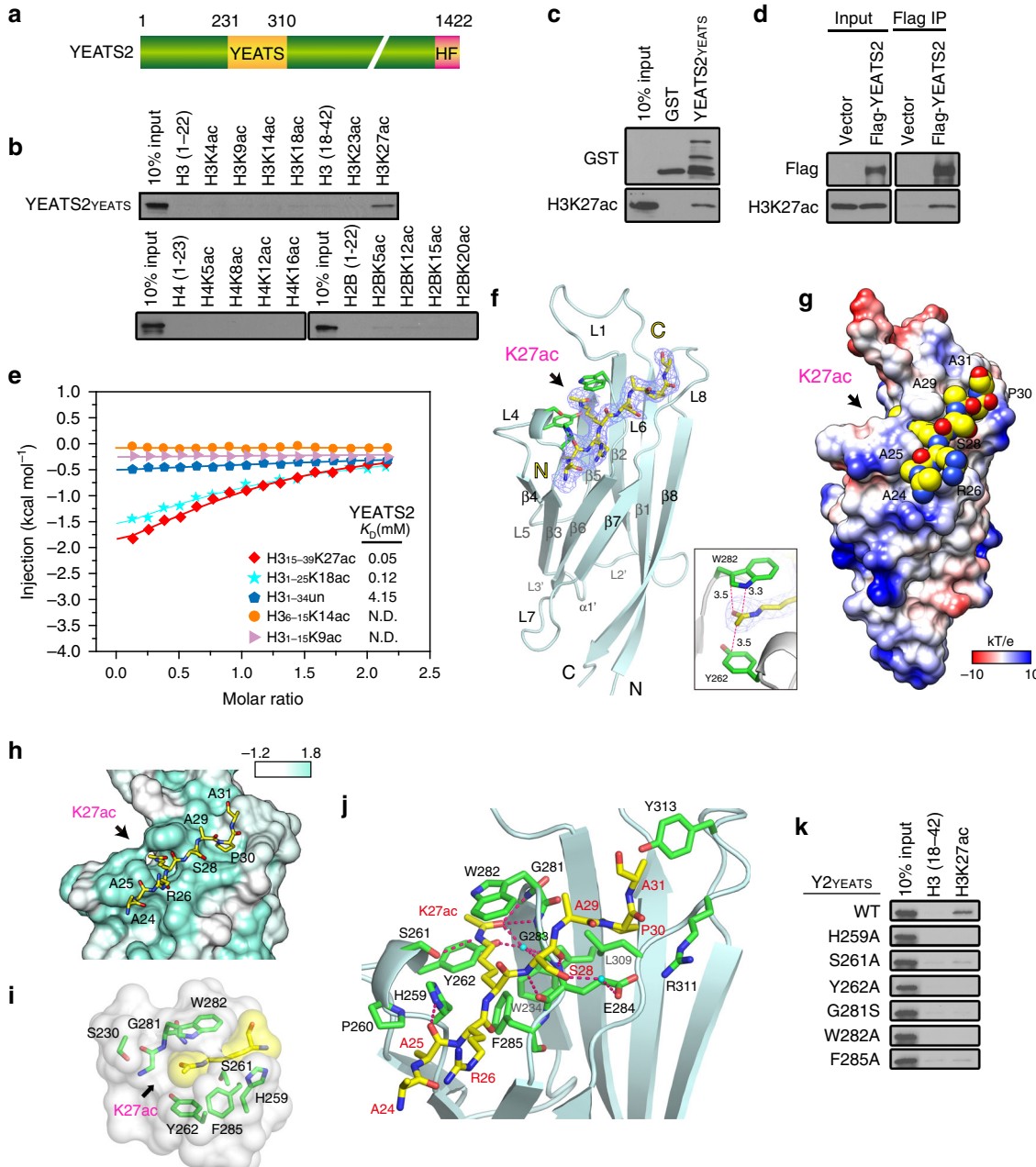

**Fig. 3** The YEATS domain of YEATS2 recognizes H3K27ac. **a** Schematic representation of YEATS2 protein structure. The amino acid numbers of the YEATS domains and the full-length protein are shown. HF: histone fold domain. **b** Western blot analysis of histone peptide pulldowns of GST-YEATS2 YEATS domain and the indicated biotinylated peptides. **c** Western blot analysis of histone pulldowns of GST-YEATS2 YEATS domain or GST and calf thymus histones. The arrow indicates the band of the GST-YEATS YEATS2 domain. **d** Western blot analysis of Flag co-IP in 293 T cells transfected with Flag-YEATS2 or vector control. **e** ITC fitting curves of YEATS2 YEATS titrated with H3$_{15-39}$K27ac, H3$_{1-15}$K18ac, unmodified H3$_{1-34}$K27, H3$_{6-15}$K14ac, and H3$_{1-15}$K9ac peptides. **f** Overall structure of YEATS2 (aa201–332) bound to the H3$_{24-31}$K27ac peptide in ribbon view. YEATS2 (pale cyan) is shown as ribbons, and the histone H3 peptide (yellow) is depicted as sticks. Purple mesh: $F_o$–$F_c$ omit map around H3$_{24-31}$K27ac peptide contoured at 1.8 σ level. Bottom right, close-up view of the Kac-sandwiching pocket; interplanar distances are labeled in the unit of angstrom. **g** YEATS2–H3K27ac space-filling-surface view color-coded by electrostatic potential ranging from −10 to 10 kT/e. **h** Conservation mapping around the H3-binding surface in YEATS2. White and cyan colors indicate low (<0.25) and high (1.0) sequence conservation, respectively. The H3K27ac peptide is shown in yellow stick. **i** Close-up view of the K27ac-binding pocket of the YEATS2 YEATS domain. The pocket is displayed as semi-transparent surface with key residues shown as green sticks. K27ac is depicted in both yellow stick and space-filling sphere modes. **j** Hydrogen bonding network between H3K27ac peptide and YEATS2. Hydrogen bonds are shown as pink dashes. Key residues of YEATS2 are depicted as green sticks and labeled black; the H3 peptide is shown as yellow sticks and labeled red. **k** Western blot analysis of peptide pulldowns of WT YEATS2 YEATS domain or the indicated point mutants with the H3K27ac peptide

**Table 1 Data collection and refinement statistics**

|  | YEATS$_{YEATS2}$–H3$_{24–31}$K27ac |
|---|---|
| *Data collection* |  |
| Space group | I422 |
| Cell dimensions |  |
| $a, b, c$ (Å) | 72.8, 72.8, 125.2 |
| $\alpha, \beta, \gamma$ (°) | 90, 90, 90 |
| Wavelength (Å) | 0.9791 |
| Resolution (Å) | 50–2.7 (2.79–2.70)* |
| $R_{merge}$ (%) | 13.3 (85.9) |
| $I/\sigma I$ | 17.53 (2.71) |
| Completeness (%) | 99.5 (95.9) |
| Redundancy | 3.8 (3.6) |
|  |  |
| *Refinement (F > 0)* |  |
| Resolution (Å) | 39.8–2.7 |
| No. of reflections (test set) | 4915 (467) |
| $R_{work}/R_{free}$ (%) | 23.0/26.3 |
| No. of atoms |  |
| Protein | 1113 |
| Ligand | 56 |
| Water | 13 |
| B-factors (Å$^2$) |  |
| Protein | 51.7 |
| Ligand | 41.7 |
| Water | 44.4 |
| R.m.s. deviations |  |
| Bond lengths (Å) | 0.005 |
| Bond angles (°) | 1.08 |

*Values in parentheses are for the highest-resolution shell

calorimetry (ITC) analysis revealed dissociation constant ($K_D$) of 0.05 mM for the YEATS domain to the H3K27ac peptide, 0.12 mM to the H3K18ac peptide, and weak or no binding to the H3K9ac, H3K14ac, or unmodified histone peptides (Fig. 3e and Supplementary Table 1).

To decipher the underlying molecular basis for recognition of histone acetylation by the YEATS domain of YEATS2, we crystallized the YEATS domain (aa 201–332) bound to the H3$_{24–31}$K27ac peptide and solved the co-crystal structure at 2.7 Å (Table 1). The overall structure of the YEATS domain adopts an immunoglobin β-sandwich fold between eight antiparallel β strands (Fig. 3f). We modeled all 132 residues of YEATS2 YEATS domain and traced the "A24-A25-R26-K27ac-S28-A29-P30-A31" residues of the H3$_{24–31}$K27ac peptide according to the electron density map. YEATS2 uses an aromatic sandwich cage for Kac recognition with the acetylamide group of Kac clamped by aromatic residues Y262 and W282 (Fig. 3g). The histone peptide-binding surface of YEATS2 formed by loops L3, L5, and L7 (corresponding to loops L4, L6, and L8 of AF9) is less negative compared to that of the AF9 YEATS domain[9], which may partly account for the relatively weak histone-binding activity observed for YEATS2.

Residue conservation analysis among YEATS2 YEATS orthologs in various species reveals strict conservation of the crucial amino acids that compose the H3K27ac-binding pocket (Fig. 3h, i and Supplementary Fig. 4a). In the complex structure of YEATS2–K27ac, the H3 peptide is stapled into the YEATS domain in an opposite N-to-C orientation compared to that of AF9 (Supplementary Fig. 4b, c). In the complex structure of AF9-K9ac, the N-terminal motif "K4-Q5-T6-A7-R8" of H3 contributes to binding whereas in the complex structure of YEATS2–K27ac, the C-terminal motif "S28-A29-P30-A31" of H3 participates in YEATS2 recognition. Notably, H3P30 at +3 position is anchored at a hydrophobic pocket of YEATS2 (Fig. 3j), which promotes

proper registration of H3K27ac. A recognition signature of "Kac-X-X-Pro" is unique to H3K27ac but not H3K9ac despite that both sites share a consensus "A-R-Kac-S" motif, which explains the binding specificity of YEATS2 to H3K27ac. The selectivity of the YEATS domain of YEATS2 was further validated by NMR experiments. [1]H, [15]N heteronuclear single-quantum coherence (HSQC) spectra of the uniformly [15]N-labeled YEATS domain showed global chemical shift perturbations upon gradual addition of the H3K27ac peptide, while H3K9ac peptide failed to induce significant chemical shift changes in the protein (Supplementary Fig. 4d). Together, these results demonstrate YEATS2 is a histone H3K27 acetylation reader.

The HSQC results indicate that the residues surrounding K27ac in the H3 peptide likely also contribute to the interaction between H3 and the YEATS2 YEATS domain. Analysis of the peptide–protein interaction using LIGPLOT program also revealed that the H3$_{24–31}$K27ac peptide is stabilized by a hydrogen bonding network and hydrophobic interactions involving a number of residues including H259, S261, Y262, W282, G283, E284, F285, and Y313 (Supplementary Fig. 4e). Indeed, alanine mutation of the sandwich pocket residues completely disrupted the binding, highlighting their essential role for H3K27ac recognition (Fig. 3k; Supplementary Fig. 4f and Supplementary Table 1). Besides, a 2-fold drop of histone P30A mutation demonstrated the requirement of flanking amino acids of H3K27 in mediating the YEATS2–H3K27ac interaction.

**The ATAC complex co-localizes with H3K27ac and H3K9ac.** The in vitro binding and structural data prompted us to determine whether YEATS2 co-localizes with acetylated histone H3K27 in cells. Extensive attempts to determine YEATS2 genomic distribution using commercial YEATS2 antibodies and tagged ectopic YEATS2 failed. Since YEATS2 is a stoichiometric component of the ATAC complex[12, 13, 15], we performed chromatin immunoprecipitation (ChIP) experiments followed by high-throughput sequencing (ChIP-seq) using a validated ChIP-seq grade antibody raised against another ATAC-specific subunit, ZZZ3[18] to represent the genome-wide distribution of the ATAC complex.

High-throughput sequencing of ZZZ3 ChIP experiments performed in duplicate identified 949 confident ZZZ3-bound peaks in H1299 cells (Supplementary Data 4). Notably, the majority of ZZZ3 peaks resided within promoter regions (82.0%), and only small fractions localized in the transcribed regions (7.9%) or intergenic regions (10.1%) likely enhancers (Fig. 4a). As YEATS2 specifically recognizes H3K27ac and the ATAC complex modifies H3K9ac, we also performed H3K27ac and H3K9ac ChIP-seq, which revealed 26,594 and 22,395 peaks, respectively (Supplementary Data 5, 6). ZZZ3 was highly co-localized with acetylated histone H3; more than 90% of the ZZZ3-binding sites were also co-occupied by both H3K27ac and H3K9ac (Fig. 4b). The heatmap and average distribution of all ZZZ3 ChIP-seq peaks across transcription units revealed a strong enrichment at regions ±1 kb of transcription start sites, largely overlapping with the genomic distribution of H3K27ac and H3K9ac (Fig. 4c, d). Genome browser views of the ChIP-seq signals of individual ZZZ3-bound genes and ChIP experiments followed by quantitative real-time PCR (ChIP-qPCR) analysis further confirmed the co-localization of ZZZ3 with H3K27ac and H3K9ac in gene promoters (Fig. 4e, f).

ZZZ3 occupies enhancer regions in GM12878 lymphoblast cells and HeLa cells[18]. To determine whether ZZZ3 also binds to enhancers in lung cancer cells, we further performed ChIP-seq to assess the genome-wide distribution of chromatin marks known to be associated with active promoters (H3K4me3) or with

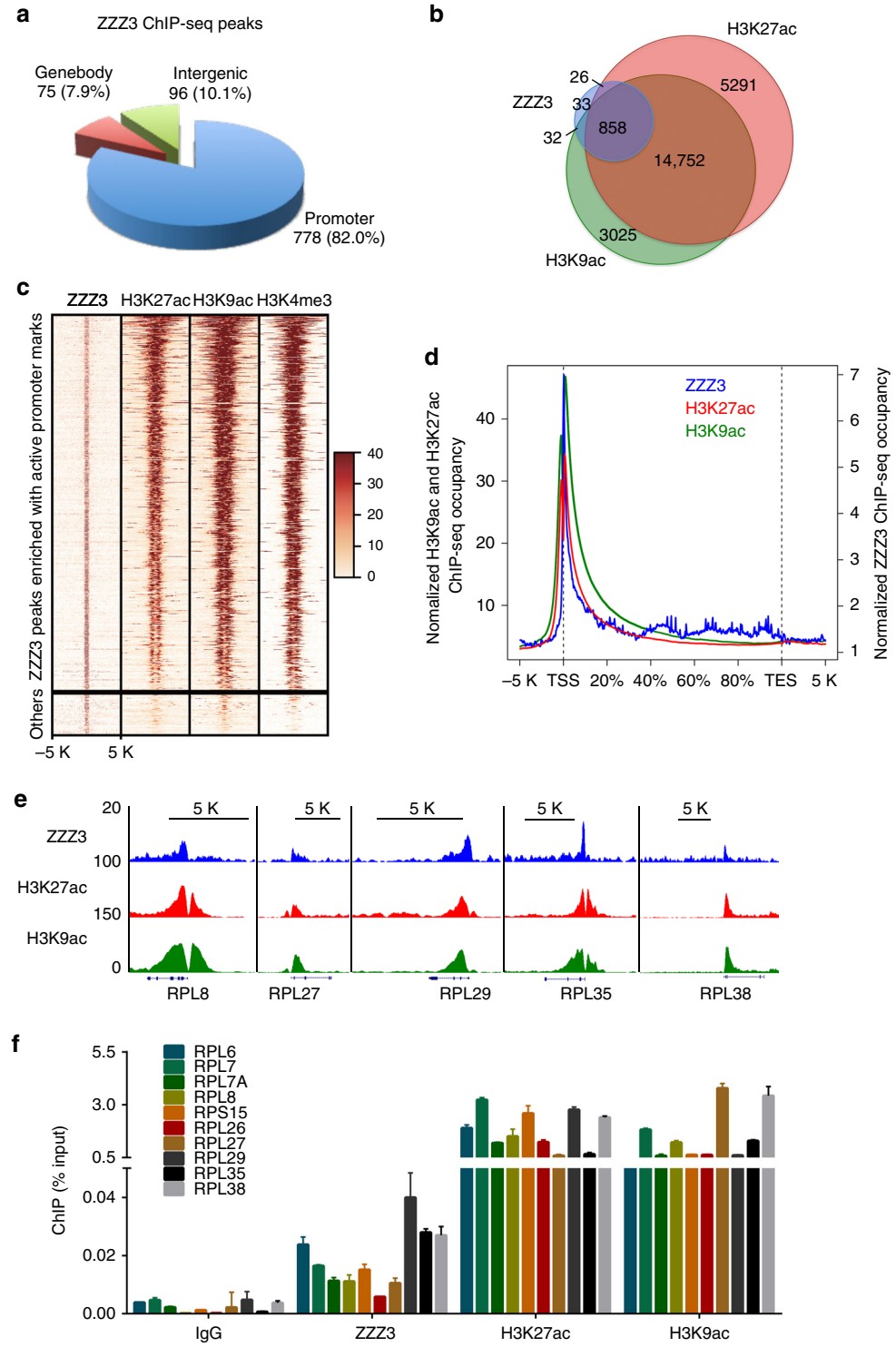

**Fig. 4** The ATAC complex co-localizes with promoter H3K27ac genome-wide. **a** Genomic distribution of ZZZ3 ChIP-seq peaks in H1299 cells. The peaks are enriched in the promoter regions (transcription start site ±3kb). $p < 2.2 \times 10^{-16}$ (binomial test). **b** Venn diagram showing the overlap of ZZZ3 (blue), H3K27ac (red) and H3K9ac (green) occupied peaks. $p < 1.79 \times 10^{-63}$ (Super exact test). **c** Heatmaps of normalized density of ZZZ3, H3K27ac, H3K9ac and H3K4me3 ChIP-seq tags centered on ZZZ3-binding peaks in a ±5 kb window. The color key represents the signal density, where darker red represents higher ChIP-Seq signal. **d** Average genome-wide occupancies of ZZZ3 (blue), H3K27ac (red) and H3K9ac (green) along the transcription unit. The gene body length is normalized by percentage from the TSS to transcription termination site (TES). 5 kb regions upstream of TSS and 5 kb regions downstream of TES are also included. **e** Genome-browser view of the ZZZ3-ChIP-seq (blue), H3K27ac-ChIP-seq (red), and H3K9acChIP-seq (green) peaks on the indicated ribosomal protein genes. **f** qPCR analysis of ZZZ3, H3K27ac and H3K9ac ChIP in the promoters of representative ribosomal protein genes. IgG was used as a negative control. Error bars indicate S.E.M. of three biological repeats

enhancers (H3K4me1) in H1299 cells. The majority of the ZZZ3-binding sites were enriched for H3K4me3, H3K9ac and H3K27ac (Fig. 4c), whereas very few ZZZ3-binding sites overlapped with non-promoter H3K4me1, a mark of enhancers (Supplementary Fig. 5a). Taken together, these results indicate that the ATAC complex co-localizes with acetylated histone H3 mainly on active promoters in H1299 cells.

**YEATS2 is required for ATAC-dependent maintenance of H3K9ac.** GCN5 and PCAF in the ATAC and SAGA complexes acetylate histone mainly on H3K9 and H3K14 to promote gene activation[15, 16, 23]. However, it still remains unknown how the ATAC complex is recruited to specific chromatin loci that are distinct from SAGA-bound regions. Because YEATS2 is an ATAC-specific subunit and binds to H3K27ac, we hypothesized that YEATS2 recruits the ATAC complex to H3K27ac-enriched target genes to promote active transcription via maintaining promoter histone H3K9/H3K14 acetylation levels. If this hypothesis is correct, depletion of YEATS2 should lead to dissociation of the ATAC complex from chromatin and reduced histone H3K9 and/or H3K14 acetylation levels. Indeed, immunoblot analysis of total histones in YEATS2 KD H1299 cells and A549 cells revealed a marked reduction in H3K9ac levels, whereas only minor or no change in H3K14ac or H4K16ac levels (Fig. 5a), suggesting that YEATS2 is required for maintaining global H3K9ac levels. Interestingly, although GCN5 and PCAF are not reported as dominate HAT enzymes for H3K27ac, we also observed marked reduction of H3K27ac levels upon YEATS2 KD (Fig. 5a). Nevertheless, stability of the ATAC complex components was not affected by YEATS2 KD, neither the HDAC1 (Supplementary Fig. 5b). Next we asked whether YEATS2 is required for ATAC-dependent histone H3K9ac on target gene promoters. To this end, we performed H3K9ac ChIP-seq in both control and YEATS2 KD cells. Averaged H3K9ac ChIP-seq signals revealed a moderate reduction of H3K9ac on the promoters of ZZZ3-occupied genes, whereas H3K9ac levels on non-ZZZ3-bound genes (others) remained largely unaffected (Fig. 5b). Consistent with the Western blot results (Fig. 5a), we also observed modest reduction of H3K27ac levels on ZZZ3-occupied genes (Supplementary Fig. 5c).

Comparison of the dysregulated genes by YEATS2 KD and the ZZZ3-occupied genes suggested that only small number genes were direct targets of the YEATS2/ATAC complex (Supplementary Fig. 5d), among which with 39 downregulated direct target genes enriched in the pathway of ribosome and in total only 10 upregulated genes enriched in two pathways (Supplementary Fig. 5e, f and Supplementary Data 7). Genome browser views of H3K9ac ChIP-seq signals and ChIP-qPCR analysis in control and YEATS2 KD cells demonstrated reduction of H3K9ac levels on the downregulated ribosomal protein-encoding genes in YEATS2 KD cells (Fig. 5c, d), whereas little or no changes in H3K9ac levels on the 10 upregulated vesicular transport or lysosome genes (Supplementary Fig. 5g). To determine whether the reduction of H3K9ac levels on the ribosomal proteins genes upon YEATS2 depletion is due to the dissociation of the ATAC complex from chromatin, we performed ZZZ3 ChIP-seq and ChIP-qPCR analyses in YEATS2 KD cells. Concurrent with changes in H3K9ac levels, we observed reduced ZZZ3 occupancy on individual ribosomal protein-encoding genes (Fig. 5c, e) as well as the averaged ChIP-seq signals of all ZZZ3-bound genes in YEATS2 KD cells (Fig. 5f). Again, in contrast, no significant changes in ZZZ3 occupancy were observed on the upregulated genes in YEATS2 KD cells (Supplementary Fig. 5h), indicating that the upregulated genes in YEATS2 KD cells are likely not direct targets of the ATAC complex. Taken together, these results indicate that YEATS2 is required for the recruitment of ATAC complex to promoters and for ATAC-dependent maintenance of histone H3K9 acetylation on the ribosomal protein-encoding genes.

**The YEATS2 YEATS domain is required for tumor cell survival.** Next, we asked whether the recognition of H3 acetylation by the YEATS domain is required for the function of YEATS2 in chromatin and transcriptional regulation. To address this question, we performed "rescue" experiments by ectopically expressing shRNA-resistant WT YEATS2 or the acetylation-binding deficient mutants (Y262A and W282A) in YEATS2-depleted H1299 cells (Supplementary Fig. 6a). We first assessed H3K9ac and the expression levels of target genes, and we found that depletion of endogenous YEATS2 reduced H3K9ac and the expression level of ribosomal protein-encoding genes (Fig. 6a, b). Importantly, ectopic expression of WT YEATS2, but not the Y262A and W282A mutants, in the YEATS2-depleted cells restored H3K9ac on target gene promoters to levels comparable to those in the control cells (Fig. 6a). Consistently, WT YEATS2, but not the H3 acetylation-binding deficient mutants, rescued target gene expression in YEATS2 KD cells (Fig. 6b).

We then sought to determine whether the YEATS domain is required for YEATS2 function in regulating cell growth and survival. We performed cell proliferation and colony formation assays using the reconstitution system in which ectopic WT or mutant YEATS2 was reintroduced to the YEATS2 KD cells. Consistent with the target gene expression patterns, ectopic expression of WT YEATS2 restored cell proliferation and colony forming capability of the YEATS2-depleted cells, whereas the Y262A and W282A mutants did not (Fig. 6c, d). Furthermore, in in vitro soft agar colony formation assays and in vivo xenograft assays, depletion of YEATS2-suppressed tumor growth. Importantly, WT YEATS2, but not the H3 acetylation-binding deficient mutants, restored the transformation capability of the YEATS2 KD H1299 cells in vitro and tumor growth in mice (Fig. 6e, f and Supplementary Fig. 6b). Taken together, our results suggest a model wherein YEATS2 recognizes histone H3 acetylation and recruits the ATAC complex to chromatin, which in turn maintains an open, acetylated chromatin environment to promote expression of genes essential for cancer cell proliferation, survival and tumorigenesis (Fig. 6g).

## Discussion
Previously we identified the AF9/ENL YEATS domain as a histone acetylation reader module[9]. In addition to AF9/ENL, the two functional paralogs that associate with the super elongation complex or the DOT1L complex[24, 25], humans have two other YEATS domain proteins, YEATS2 and YEATS4/GAS41, which are components of the ATAC HAT complex and TIP60/SRCAP chromatin-remodeling complexes, respectively[12, 16, 26, 27]. Our biochemical and structural studies reveal that, different from the AF9 YEATS domain that recognizes acetylation on H3K9, K18 and K27, the YEATS domain of YEATS2 shows certain specificity, with H3K27ac as the best binding substrate whereas no detectable binding was observed to H3K9ac or H3K14ac. Nevertheless, the YEATS domain of YEATS2 utilizes aromatic residues conserved among all YEATS domains forming a Ser/Thr-lined sandwiching cage for encapsulation of the acetyl moiety. Together with our unpublished data of GAS41, our results demonstrate that recognition of histone acetylation is a common feature of YEATS domains involving distinct chromatin-remodeling or histone-modifying complexes.

The hydrophobic pockets of all known YEATS domains are "open-ended", enabling recognition of other types of acylation

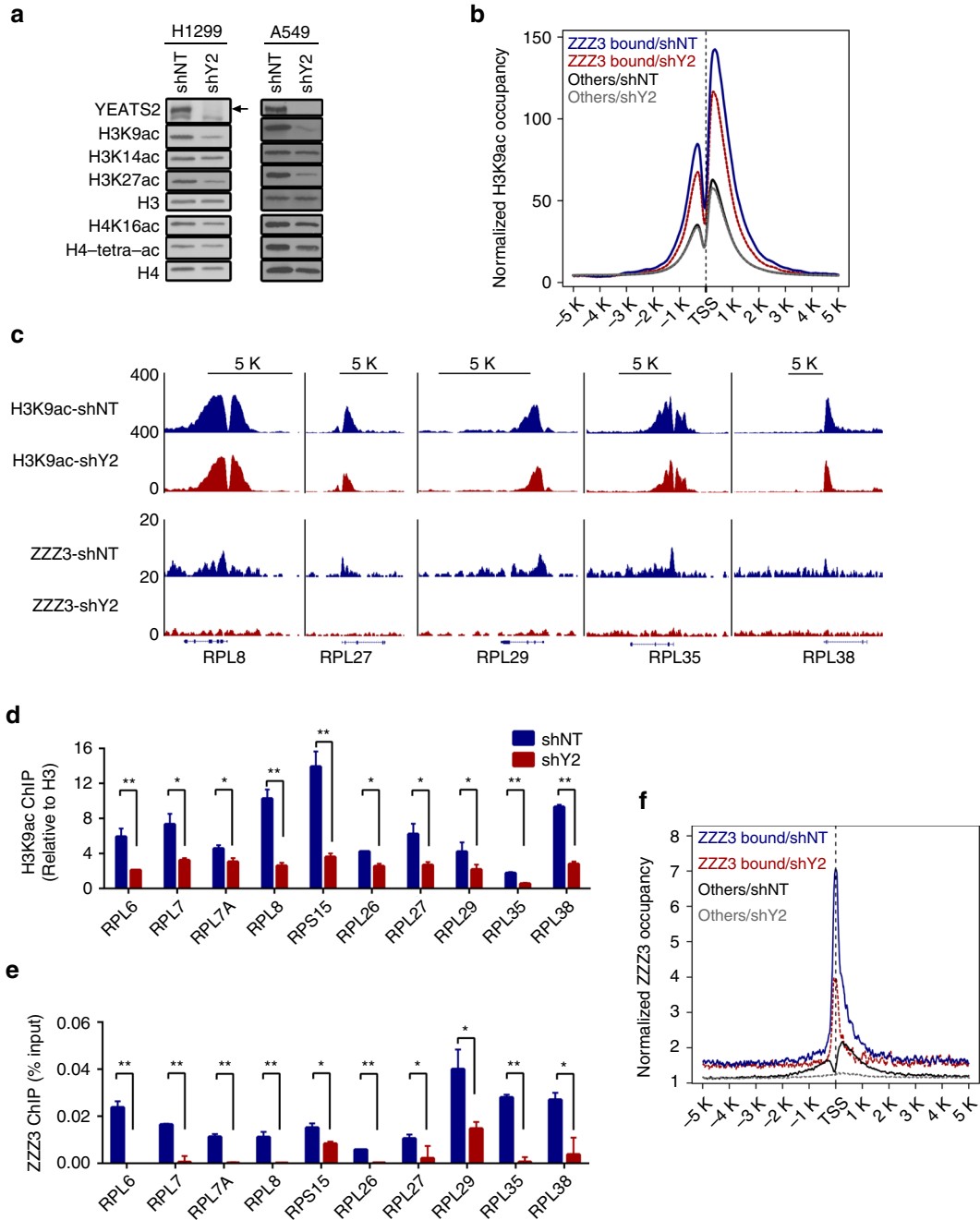

**Fig. 5** YEATS2 is required for ATAC-dependent maintenance of histone H3K9ac on the ribosomal protein genes. **a** Western blot analysis of YEATS2 and H3 and H4 acetylation in control (shNT) and YEATS2 KD (shY2) cells. H3 and H4 were used as a loading control. The arrow indicates the band of YEATS2 protein. **b** Average genome-wide H3K9ac occupancy on the promoter (5 kb±TSS) of the ZZZ3-bound genes or non-ZZZ3-bound genes (others) in control (shNT) and YEATS2 KD (shY2) H1299 cells. **c** Genome-browser view of the H3K9ac and ZZZ3 ChIP-seq peaks on the indicated ribosomal protein genes in cells as in (**b**). **d** qPCR analysis of H3K9ac ChIP of the indicated ribosomal protein genes in cells as in (**b**). **e** qPCR analysis of ZZZ3 ChIP of the indicated ribosomal protein genes in cells as in (**b**). **f** Average ZZZ3 occupancy on the promoter (5 kb±TSS) of the ZZZ3-bound genes or non-ZZZ3-bound genes (others) in control (shNT) and YEATS2 KD (shY2) H1299 cells. In **d** and **e**, error bars indicate S.E.M. of at least three biological replicates. *$p < 0.05$; **$p < 0.01$ (Student's $t$-test)

with longer chains. Indeed, recently we found that AF9, YEATS2, and yeast Taf14 proteins are capable of binding to a repertoire of histone acylations, with slightly higher affinities to crotonylation[28–30]. Crotonylation and other types of acylations, such as propionylation, butyrylation, and β-hydroxybutyrylation, have been detected on histones in a variety of species[31–33]. Similar to acetylation, these modifications on histones are associated with active transcription. However, the abundances of these newly identified histone modifications are at levels of orders of magnitude lower than that of acetylation in cells[33, 34], raising the question how cells discriminate these chemically closely related modifications. In the current study, we failed to detect H3K27cr ChIP-seq signals, likely due to the low abundance of this mark in H1299 cells under normal growth conditions. Interestingly,

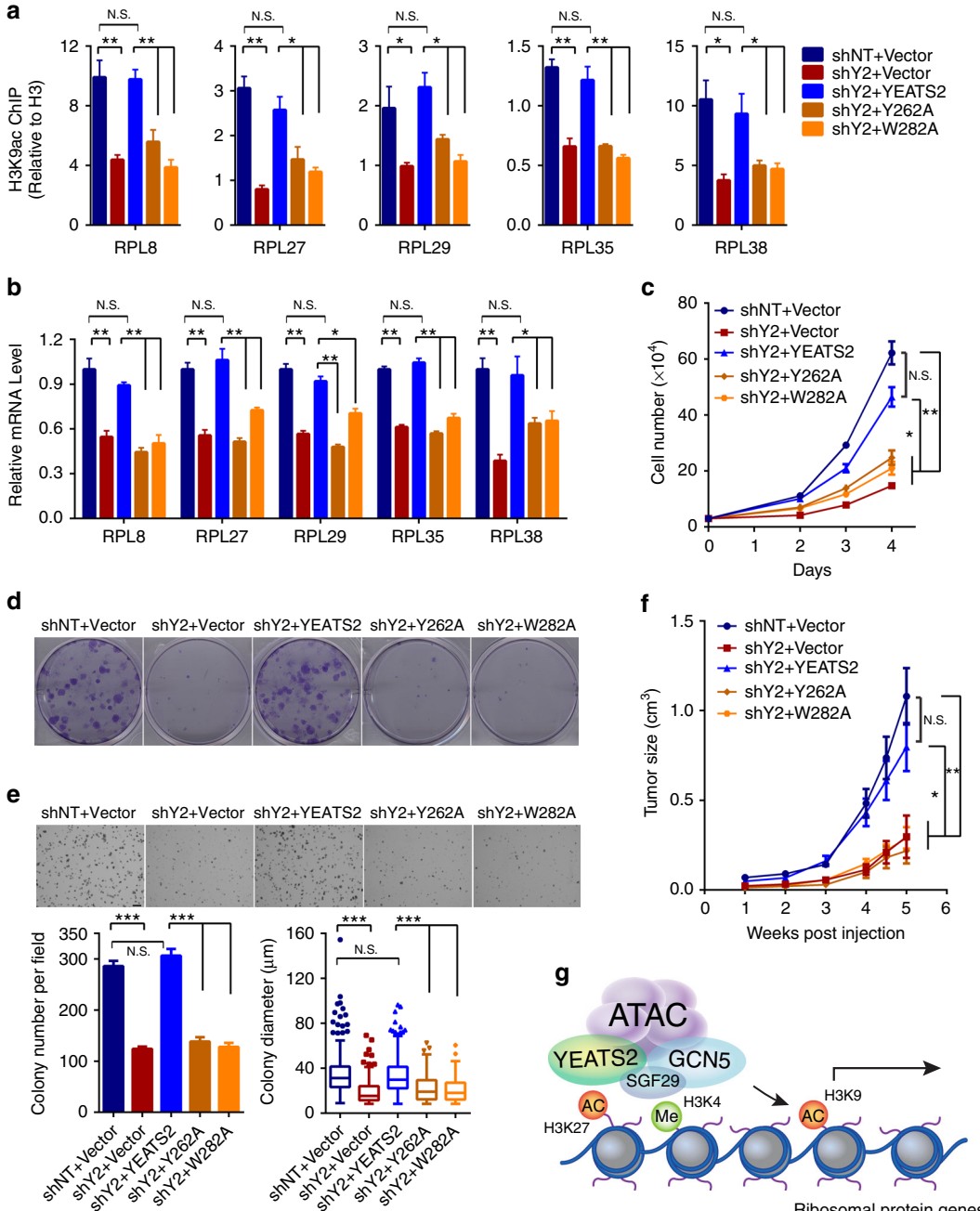

**Fig. 6** The YEATS domain of YEATS2 is required for ATAC-dependent ribosomal protein gene expression and tumor cell survival. **a** qPCR analysis of H3K9ac ChIP in the promoters of the indicated ribosomal protein genes in control (shNT) and YEATS2 KD (shY2) H1299 cells ectopically expressing shRNA-resistant WT YEATS2 or the indicated mutants. **b** qRT-PCR analysis of the expression of ribosomal protein genes in cells as in (**a**). In **a** and **b**, error bars indicate S.E.M. of at least three biological replicates. N.S. not significant; *$p < 0.05$; **$p < 0.01$ (Student's $t$-test). **c** Cell proliferation assay of cells as in (**a**). Cells (mean ± S.E.M., $n = 3$) were counted for 4 days after seeding. Error bars represent the S.E.M. N.S.; *$p < 0.05$; **$p < 0.01$ (Student's $t$-test). **d** Clonogenic assay of cells as in (**a**). Colonies were stained and photographed 7 days after seeding. **e** Anchorage-independent growth assay of cells as in (**a**). Cells (mean ± S.E.M., $n = 4$–6) were stained and photographed 3 weeks after seeding. Colony numbers (bottom left) and diameters (bottom right) were measured using ImageJ software. Error bars represent the S.E.M. Scale bar, 200 μm. N.S.; ***$p < 0.001$ (Student's $t$-test). **f** Volumes of tumors (mean ± S.E.M., $n = 10$) of the H1299 cells as in (**a**) subcutaneously transplanted into immunodeficient nude mice. Tumors were monitored for 5 weeks after transplantation. N.S.; *$p < 0.05$; **$p < 0.01$ (Student's $t$-test). **g** Working model: the YEATS2 subunit of the ATAC complex recognizes H3K27ac through its YEATS domain and stabilizes the ATAC complex at target promoter regions to maintain local histone acetylation and gene expression, which are essential for cell growth and survival. Note that additional reader modules, such as the SGF29 double Tudor domains that bind to H3K4me3, also contribute to chromatin association of the ATAC complex

several recent studies suggest that these alternative acylations likely play a prominent role controlling gene expression at specific developmental stages or in responses to certain stresses[33–35]. Nevertheless, given the preponderant abundance, histone acetylation likely plays a dominant role in epigenetic regulation of gene expression under most circumstances.

Acetylation on histone H3 K9 and K14 are known as marks for active transcription[36]. In mammals, acetylation on these two residues is predominantly deposited by GCN5/PCAF present in SAGA or ATAC, two HAT complexes with non-overlapping functions[14, 15, 20]. SAGA is principally found at gene promoters in general whereas ATAC occupies both promoter and enhancer regions in GM12878 lymphoblast cells and HeLa cells[18]. However, to our surprise, we did not observe any significant enhancer occupancy of the ATAC complex in the NSCLC H1299 cells, indicating that the enhancer occupancy of the ATAC complex in different cell types is likely context-dependent. Moreover, other reader modules within the complexes may also contribute the binding specificity. In line with this speculation, SGF29, a shared subunit of the SAGA and ATAC complexes, has been shown to recognize H3K4me3 through its double Tudor domains[37, 38], and the BRDs of GCN5 and PCAF bind to acetylated histones[39–41]. Therefore, YEATS2 likely cooperates with SGF29, GCN5/PCAF, and possibly some other yet unknown reader(s) to form a "reader module" within the ATAC complex facilitating chromatin recruitment of the complex. As both SGF29 and GCN5/PCAF are shared subunits of SAGA and ATAC, the ATAC-specific YEATS2 and promoter H3K27ac may account for, at least in part, the differential distributions of SAGA and ATAC in promoters. Furthermore, given that H3K27ac, but not H3K4me3, is enriched in active enhancers, we speculate that the enhancer occupancy of the ATAC complex in other cells likely depends on the functionality of the YEATS2 YEATS domain rather than the SGF29 Tudor domains.

The SAGA complex is known to play an essential role for both normal and neoplastic development[20]. Components of the SAGA complex directly interact with the Myc oncoprotein and a plethora of transcription factors regulating gene expression involving in diverse processes[42–44]. In contrast, little is known about the pathways that ATAC complex is involved. In *Drosophila*, the ATAC complex serves as a transcriptional cofactor for c-Jun-regulating JNK target genes[19], and in mammals, ATAC activates gene expression during stress responses[15, 45]. In the current study, we find that the ATAC complex also transcriptionally regulates a large number of essential genes including the ribosomal protein-encoding genes. The *YEATS2* gene is frequently amplified in NSCLC, especially the squamous sub-type. Knockdown of YEATS2 dampens the expression of 49 out of the total 79 ribosomal protein genes and suppresses the growth and survival of a panel of lung cancer cells, suggesting a growth dependency of NSCLC on YEATS2, and possibly the ATAC complex. Interestingly, Myc is known to regulate the expression of genes encoding ribosomal proteins and several other components of the protein synthetic machinery[46, 47]. It is of interest to determine in future studies whether the ATAC complex also directly interacts with the Myc oncoprotein, or whether ATAC cooperates with the SAGA complex in transcriptionally regulating the protein synthetic machinery. Taken together, the identifications of YEATS2 as a histone acetylation reader and a candidate oncogene amplified in NSCLC suggest that the YEATS domain may provide an attractive therapeutic target for treatment.

## Methods

**Materials**. Human YEATS2 cDNA (NCBI Gene ID 55789) was cloned into pENTR3C, and subsequently cloned into pCDH destination vectors using Gateway techniques (Invitrogen). The cDNAs encoding the YEATS domains of human YEATS2 (aa 184–449 or aa201–332) were cloned into the pGEX-6P1 vectors (Novagen). Point mutations were generated using a site-directed mutagenesis kit (Stratagene). Histone peptides bearing different modifications were synthesized at the W.M. Keck Facility at Yale University or CPC Scientific Inc. Anti-histone antibodies including anti-H3 (Ab1791, WB 1:20000), anti-H3K9ac (Ab32129, WB 1:1000), anti-H3K14ac (Ab52946, WB 1:1000), anti-H3K27ac (Ab4729, WB 1:1000), anti-H4 (Ab731, WB 1:5000), and anti-HDAC1(ab19845, WB 1:1000) antibodies were obtained from Abcam; anti-H3K9ac (61251) and anti-H4K16ac (39167 WB 1:1000) from Active Motif; anti-H4 tetra-acetyl antibody (06-598, WB 1:5000) from Millipore; anti-GCN5 (sc-20698, WB 1:2000), anti-PCAF (sc-13124, WB 1:200), anti-ADA3 (sc-98821, WB 1:1000), anti-ATAC2 (sc-398475, WB 1:1000), and anti-GST (sc-459, WB 1:1000) antibodies from Santa Cruz; anti-actin (A1978, WB 1:5000), anti-ZZZ3 (SAB4501106, WB 1:1000), and anti-tubulin (T8328, WB 1:5000) antibodies from Sigma; and anti-YEATS2 (24717-1-AP, WB 1:1000) antibody from ProteinTech. SGF29 antibody (WB 1:1000) and ChIP-seq grade ZZZ3 antibody were made in Dr. Laszlo Tora's laboratory[18]. pLKO shRNA constructs were purchased from Sigma. The shRNA sequences were: YEATS2#1: GCACAGAAACTGACTTCTTTA; YEATS2#2: TCAAAGAACTTGGTCATAAAT.

**Protein production**. The YEATS2 domain encompassing residues 201–332 of human YEATS2 was cloned into a pSUMOH10 vector (an in house modified vector based on pET28b) containing an N-terminal 10×His-SUMO tag. The recombinant YEATS2$_{201-332}$ was overexpressed in *Escherichia coli* BL21 (DE3). After overnight induction by 0.4 mM isopropyl β-D-thiogalactoside at 16 °C in TB medium, cells were collected and suspended in buffer: 20 mM Tris, pH 7.5, 0.5 M sodium citrate, 5% glycerol, 1 mM phenylmethylsulfonyl fluoride, and 20 mM imidazole. After cell lysis and centrifugation, the recombinant protein was purified to homogeneity over HisTrap, and the 10xHis-SUMO tag was cleaved by ULP1 overnight at 4 °C then removed by reloaded onto the HisTrap column. The free YEATS2$_{201-332}$ protein was finally polished by size-exclusion chromatography on a Superdex G75 column (GE Healthcare) in elution buffer: 20 mM Tris, pH 7.5, 0.5 M sodium citrate, 5% glycerol, and 2 mM β-mercaptoethanol. All YEATS2$_{201-332}$ mutants were purified in essentially the same procedures as the wild-type protein. All mutant proteins were expressed and purified essentially the same as WT YEATS2 YEATS. For NMR titrations, the YEATS domain (aa 201–350) of YEATS2 was expressed in BL21(DE3) RIL cells as a GST-fusion protein in minimal media supplemented with $^{15}$NH$_4$Cl (Sigma). Cells were pelleted via centrifugation, flash-frozen in liquid nitrogen, and lysed by sonication. Cell lysate was centrifuged, and the supernatant was incubated with glutathione Sepharose 4B beads (GE Healthcare). The GST tag was cleaved with Prescission protease. The $^{15}$N-labeled YEATS2 YEATS domain was concentrated in 1× PBS (pH 6.8) buffer supplemented with 100 mM KCl before NMR experiments.

**Crystallization and structure determination**. For YEATS2$_{201-332}$–H3K27ac complex, the sample was prepared by direct mixing protein with a H3$_{24-31}$K27ac (ATKAARKacSAPA) in a 1:10 molar ratio. The crystals were generated by sitting drop vapor diffusion method. Briefly, protein droplets containing 1 µl of YEATS2$_{201-332}$–H3K27ac (6.5 mg/ml) were mixed with 1 µl of reservoir solution (0.2 M lithium sulfate, 2.0 M ammonium sulfate, 0.1 M 3-(cyclohexylamino)-1-propanesulfonic acid (CAPS), pH 10.5) and incubated in a closed 48-well plate at 18 °C for 3 days. The crystals were then collected and briefly soaked in a cryo-protectant drop composed of the reservoir solution supplemented with 30% glycerol and then flash frozen in liquid nitrogen for data collection. The diffraction data set was collected at the beamline BL17U of the Shanghai Synchrotron Radiation Facility at 0.9791 Å. All diffraction images were indexed, integrated, and merged using HKL2000[48]. The structure was determined by molecular replacement using MOLREP[49] with the AF9 complex structure (PDB ID: 4TMP) as the search model. Structural refinement was carried out using PHENIX[50], and iterative model building was performed with COOT[51]. Detailed data collection and refinement statistics are summarized in Table 1. Structural figures were created using the PYMOL (http://www.pymol.org/) or Chimera (http://www.cgl.ucsf.edu/chimera) programs.

**Isothermal titration calorimetry**. All calorimetric experiments of the wild type or mutant YEATS domain proteins were conducted at 15 °C using a MicroCal iTC200 instrument (GE Healthcare). The YEATS2$_{201-332}$ samples were dialyzed in the following buffer: 20 mM Tris 7.5, 0.5 M sodium citrate, 5% glycerol, and 2 mM β-mercaptoethanol. Protein concentration was determined by absorbance spectroscopy at 280 nm. Peptides (>95% purity) were quantified by weighing on a large scale and then aliquoted and freeze-dried for individual use. Acquired calorimetric titration curves were analyzed using Origin 7.0 (OriginLab) using the "One Set of Binding Sites" fitting model. Detailed peptide sequence information is summarized below: H3$_{15-39}$K27ac: APRKQLATKAARK(ac)SAPATGGVKKPH, H3$_{1-34}$K27un: ARTKQTARKSTGGKAPRKQLATKAARKSAPATGG, H3$_{1-15}$K9ac: ARTKQ-TARK(ac)STGGKA.

**NMR titrations**. NMR experiments were carried out on a Varian INOVA 600 MHz at 298 K. $^1$H, $^{15}$N heteronuclear single-quantum coherence (HSQC)

spectra were collected on 0.1 mM uniformly [15]N-labeled YEATS domain of YEATS2 (aa 201–350 of YEATS2) in 1× PBS (pH 6.8) supplemented with 100 mM KCl and ~8% $D_2O$ in the presence of increasing concentration of H3 peptides (H3K27ac$_{21-31}$ or H3K9ac$_{1-12}$).

**Peptide pulldown assay and GST pulldown assay.** An aliquot of 1 μg of bioti-nylated histone peptides with different modifications were incubated with 1–2 μg of GST-fused proteins in binding buffer (50 mM Tris-HCl 7.5, 250 mM NaCl, 0.1% NP-40, 1 mM phenylmethyl sulphonyl fluoride (PMSF)) at 4 °C overnight. Strep-tavidin beads (Amersham) were added to the mixture, and the mixture was incubated for 1 h with rotation. The beads were then washed three times and analyzed using SDS-PAGE and western blotting. For GST pulldown, 2 μg protein were incubated with 10 μg of calf thymus total histones (Worthington) in binding buffer (50 mM Tris-HCl 7.5, 1 M NaCl, 1% NP-40, 0.5 mM EDTA, 1 mM PMSF plus protease inhibitors (Roche)) at 4 °C overnight, followed by an additional 1 h Glutathione Sepharose beads (Amersham) incubation. The beads were then washed five times and analyzed using SDS-PAGE and western blotting.

**Protein-chromatin immunoprecipitation.** Protein-ChIP assays for detection of YEATS2-histone interactions were performed as described below[52]. Briefly, cells were crosslinked with 1% formaldehyde for 10 min and stopped with 125 mM glycine. The isolated nuclei were resuspended in nuclei lysis buffer and sonicated. The nuclei lysate was diluted in cell lysis buffer (20 mM Tris-HCl 8.0, 150 mM NaCl, 1% Triton X-100, 1 mM EDTA, 0.01% SDS, 1 mM PMSF plus protease inhibitors (Roche)). Anti-FLAG M2-conjugated agarose beads (Sigma) were incubated with the lysates overnight at 4 °C. The beads were then washed with low salt (20 mM Tris pH 8.0, 150 mM NaCl, 2 mM EDTA, 1% Triton X-100, 0.1% SDS), high salt (20 mM Tris pH 8.0, 500 mM NaCl, 2 mM EDTA, 1% Triton X-100, 0.1% SDS), and LiCl buffer (20 mM Tris pH 8.0, 250 mM LiCl, 1 mM EDTA, 1% NP-40, 1% sodium deoxycholate), and the bound proteins were eluted in SDS buffer and analyzed by western blotting. All uncropped blots are provided in Supplementary Fig. 7.

**Cell culture and RNA interference.** All cell lines were tested for mycoplasma contamination and validated by STR DNA fingerprinting performed by the MDACC CCSG-funded Characterized Cell Line Core (NCI #CA016672). Human HEK 293 T, fibroblasts WI-38 and IM-R90 (ATCC), and human ovarian cancer cell lines CaoV3 and HeyA8 (gifts from Dr. Xiongbin Lu) were maintained in DMEM (Cellgro) supplemented with 10% fetal bovine serum (Sigma). Human lung cancer cell lines H1299, A549, H1355, Ludlu-1, and H520 (gifts from Dr. J. Hey-mach) were cultured in RPMI 1640 (Cellgro) supplemented with 10% fetal bovine serum. Retroviral or lentiviral transduction was performed as described below[53]. Briefly, 293 T cells were co-transfected with pMD2.G, pPAX2 (Addgene), and pLKO shRNA or pCDH cDNA constructs. For infections, cells were incubated with viral supernatants in the presence of 8 μg/ml polybrene. After 48 h, the infected cells were selected with puromycin (2 μg/ml) for pLKO clones or blasticidin (10 μg/ml) for pCDH clones for 3–4 days before experiments.

**Real-time PCR and RNA-seq analysis.** Total RNA was extracted using an RNeasy plus kit (Qiagen) and reverse-transcribed using an iScrip reverse transcription kit (Bio-Rad). Quantitative real-time PCR (qPCR) analyses were performed as described previously using Power SYBR Green PCR Master Mix and the ABI 7500-FAST Sequence Detection System (Applied Biosystems)[53]. Gene expressions were calculated following normalization to GAPDH levels using the comparative Ct (cycle threshold) method. Statistic differences were calculated using a two-way unpaired Student's *t*-test. The primer sequences for qPCR are listed in Supple-mental Table 2.

RNA-seq samples were sequenced using the Illumina Hiseq 2500, and raw reads were mapped to the human reference genome (hg19) and transcriptome using the RNA-Seq unified mapper. Read counts for each transcript were calculated using HTseq v0.6.1 using default parameters[54]. Differential gene expression analyses were performed using the "exactTest" function in edgeR v3.0[55]. Gene Ontology analysis was performed using the DAVID Bioinformatics Resource 6.7[56]. The gene expression heatmap was generated using pheatmap package in CRAN (https://cran.r-project.org/package=pheatmap). The volcano plot was drawn by using ggplot2 package (https://cran.r-project.org/package=ggplot2) in R computing environment.

**Cell proliferation and colony formation assays.** Cell proliferations were deter-mined by counting live cells using hemocytometer cell counter or by CellTiter-Glo luminescent cell viability assay kit (Progema-G7572). For colony formation assays, H1299 cells were seeded in 6-well tissue culture plates (400 cells/well) and grown for 10–14 days. Colonies were fixed with glutaraldehyde (6.0% v/v), stained with crystal violet (0.5% w/v) and photographed.

Soft agar assays were performed as described below[53]. Briefly, cells (1 × 10[4]) were suspended in 1 ml top agar medium (culture medium supplied with 0.35% agar). The cell suspensions were then overlaid onto 1.5 ml bottom agar medium (culture medium supplied with 0.6% agar) in six-well tissue culture plates in triplicate. Fresh medium was added to plates every 3 days. On day 21, cells were

stained with 0.005% crystal violet blue and photographed. Colony numbers and colony diameters were measured using ImageJ software with size cutoff of 15 μm. Results were quantitated from six views per sample of at least three independent replicates.

**Flow cytometry cell cycle analysis.** Cells were harvested and single cell suspen-sion was prepared at 2 × 10[6] in 1 ml ice-cold PBS buffer. The cell suspension was added dropwise to 9 ml 70% ethanol for fixing. The samples were kept at least 2 h at 4 °C then washed in cold PBS twice. Then the cells were treated with 100 μg/ml RNase A in PBS for 20 mins at room temperature, followed by adding propidium iodide (PI; 50 μg/ml) for staining. The cell cycle profiling was analyzed by flow cytometry using 488 nm excitation.

**FITC Annexin V apoptosis assay.** Phosphatidylserine (PS) translocation from the inner to the outer leaflet of plasma membrane is one of the earliest apoptotic features. The binding of Annexin V to cell surface PS was detected with a com-mercially available FITC Annexin V Apoptosis Detection Kit I (BD Pharmingen 556547). Briefly, 1 × 10[5] cells were pelleted, resuspended in 100 μl of Hepes-buffered saline, and FITC-labeled annexin V and PI were added. The cells were incubated 15 min at room temperature, then the samples were transferred to ice and the sample volume brought to 0.5 ml. Analysis was done by flow cytometry within 1 h. The results were analyzed with FlowJo software. Annexin V positive cells were determined as described in the Kit by setting quadrants to separate viable cells from PI permeant cells, and non-apoptotic cells from those staining highly for the FITC-labeled Annexin V probe. Percent apoptosis was determined from the cells staining greater than the control population threshold.

**Transwell cell migration assay.** Cell migration was assayed using Transwell chambers (6.5 mm; Corning, Corning, NY, USA) with 8 μm pore membranes. The lower chamber was filled with 500 μl of 10% FBS RPMI 1640 medium. A total of 1 × 10[5] cells were suspended with 500 μl FBS-free RPMI 1640 medium and placed into the upper chamber. After 14 h, cells were fixed using 5% glutaraldehyde and stained using 0.5% crystal violet. Cells in the upper chamber were carefully removed, and cells that migrated through the membrane were assessed by pho-tography. For quantification, crystal violet was extracted by methanol and the absorbance at 540 nm was measured.

**ChIP and ChIP-seq analysis.** ChIP analysis was performed essentially as described below[53]. Briefly, cells were crosslinked with 1% formaldehyde for 10 min and stopped with 125 mM glycine. The isolated nuclei were resuspended in nuclei lysis buffer and sonicated using a Bioruptor Sonicator (Diagenode). The samples were immunoprecipitated with 2–4 μg of the appropriate antibodies overnight at 4 °C. Protein A/G beads were added and incubated for 1 h, and the immunoprecipitates were washed twice each with low salt, high salt, and LiCl buffers. Eluted DNA was reverse-crosslinked, purified using PCR purification kit (Qiagene), and analyzed by quantitative real-time PCR on the ABI 7500-FAST System using the Power SYBR Green PCR Master Mix (Applied Biosystems). Statistic differences were calculated using a two-way unpaired Student's *t*-test. The primers used for qPCR are listed in the Supplementary Table 2.

For ChIP-seq, ChIP experiments were carried out essentially the same as described above. Samples were sequenced using the Illumina Solexa Hiseq 2500. The raw reads were mapped to human reference genome NCBI 37 (hg19) by Solexa data processing pipeline, allowing up to 2 mismatches. The genome ChIP-seq profiles were generated using MACS 1.3.6[57] with only unique mapped reads. Clonal reads were automatically removed by MACS. The ChIP-seq profiles were normalized to 10,000,000 total tag numbers, and peaks were called at $p \le 1e{-}8$. The ChIP-seq heatmap was drawn by the seqplots R package (http://github.com/przemol/seqplots).

**Tumor xenograft.** All animal studies were in compliance with ethical regulations at the University of Texas MD Anderson Cancer Center. Female athymic nude mice (age 6–8 weeks) were obtained from University of Texas MD Anderson Cancer Center and animals were housed under pathogen-free conditions. Tumor xenograft assay was performed as described below[53]. Briefly, three million YEATS2 knockdown H1299 cells stably expressing control pCDH vector, wild-type YEATS2, Y262A, or W282A mutants were suspended in 100 μl of serum-free RPMI 1640 and injected subcutaneously into the mice. The growth of tumors was monitored twice a week until the largest one reached the limit of tumor burden. Tumor sizes were measured using a caliper and tumor volume was calculated according to the following equation: tumor volume (mm[3]) = (length (mm) × width[2] (mm[2])) × 0.5. Representative data were obtained from all the mice per experimental group. Statistical analyses were performed with one-way repeated-measures analysis of variance.

**Statistical analyses.** Experimental data are presented as means ± standard deviation of the mean unless stated otherwise. Statistical significance was calculated unless stated otherwise by two-tailed unpaired *t*-test on two experimental condi-tions with $p < 0.05$ considered statistically significant. Statistical significance levels

are denoted as follows: *$p < 0.05$; **$p < 0.01$; ***$p < 0.001$; ****$p < 0.0001$. No statistical methods were used to predetermine sample size. Super exact test was performed to test the significance of the Venn diagram by the R package Exact (https://cran.r-project.org/package=Exact).

**Data availability**. Structure data are deposited in the Protein Data Bank with the accession code 5XNV. The ChIP-seq and RNA-seq data are deposited in the GEO database with the accession number GSE90781.

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

## Acknowledgements

We thank J. Heymach, J. Minna, J. Kurie, M. Bedford, M.G. Lee, and X. Lu for sharing reagents. We thank the M.D. Anderson Sequencing and Microarray Facility and the Science Park Next-Generation Sequencing Facility (CPRIT RP120348) for Solexa sequencing. We thank the staff at beamline BL17U of the Shanghai Synchrotron Radiation Facility and Dr. S. Fan at Tsinghua Center for Structural Biology for their assistance in data collection and the China National Center for Protein Sciences Beijing for providing facility support. We thank B. Dennehey for editing the manuscript. This work was supported in part by grants from NIH/NCI (1R01CA204020-01), Cancer Prevention and Research Institute of Texas (RP160237 and RP140323), Welch Foundation (G1719), and Texas Tobacco Settlement to X.S., CPRIT (RP110471 and RP150292) and NIH (R01HG007538 and R01CA193466) to W.L., NIH (R01GM100907) to T.G.K., NIH (R01GM067718) to S.Y.R.D., National Natural Science Foundation of China (91519304), Major State Basic Research Development Program in China (2015CB910503), and Tsinghua University Initiative Scientific Research Program to H.L., National Postdoctoral Program for Innovative Talents (BX201600088) to D.Z., and European Research Council (ERC) (ERC-2013-340551, Birtoaction) to L.T.

## Author contributions

W.M., H.G., J.L. and D.Z. contributed equally to this work. X.S., H.L., W.L. and W.M. conceived the study. W.M. performed the biochemical and cellular studies; H.G. and D.Z. performed structural and calorimetric studies; J.L. and Y.X. performed bioinformatics analysis; S.J. performed xenograft assays, F.H.A. performed NMR titration studies; M.G. performed pathological analysis of tumors; X.W. provided technical assistance; L.T. provided ChIP-grade anti-ZZZ3 polyclonal antibody; X.S., H.L. and W.M. wrote the paper with comments from H.W., S.Y.R.D., L.T., J.L. and W.L.

## Additional information

Competing interests: W.M. is a M.D. Anderson Center for Cancer Epigenetics postdoctoral scholar. D.Z. is a postdoctoral fellow of Tsinghua-Peking Joint Center for Life Sciences. F.H.A. is an AHA postdoctoral fellow. W.L. is a fellow of the Jane Coffin Childs Memorial Fund. X.S. is a Leukaemia & Lymphoma Society Career Development Program Scholar and a R. Lee Clark Fellow and Faculty Scholar of M.D.. Anderson Cancer Center. X.S. is a Scientific Advisory Board member of EpiCypher. The remaining authors declare no competing financial interests.

