## [Peer Review File · Nature Communications]

Reviewers' comments:

Reviewer #1 (Remarks to the Author):

This work of Mi et al reports that YEATS2, a gene encoding a human YEATS domain-containing chromatin reader, is highly amplified in cancer including non-small cell lung cancer (NSCLC). Authors found YEATS2 is required for NSCLC growth and transforming potential, partly through maintaining essential gene expression programs such as ribosomal proteins to sustain cell proliferation. The YEATS domain of YEATS2 is a selective reader for H3K27ac. ChIP-seq of a YEATS2-associated complex component also reveals its binding mainly at gene promoters with H3K27ac. Lastly, authors carried out point mutagenesis at the YEATS domain to demonstrate that reading of H3K27ac is essential for recruitment of YEATS2-associated HAT complex (ATAC) to promoter, ATAC-mediated histone acetylation (H3K9ac), and cancer-promoting functions.

This work is done by the team who initially made discovery of YEATS domains as a novel "reader" class specific to histone acetylation. A novel finding of this current work is to link YEATS2 and its histone H3K27ac-selective "reader" function to cancer. This study uses an integrated approach involving cancer cell biology, structural biology and genomics profiling to delineate mechanistic details of a cancer related pathway, which was not reported before. The implication of the work is far reaching in terms of therapeutics. For example, targeting the histone acetylation reader domain of BRD4 is now widely accepted as a promising anti-cancer strategy. Therefore, the report is also timely and shall appeal to the field and readers.

Comments:

1/ Fig. 1f: author needs to state the source of data for prognosis of human lung cancer patients. Is it from TCGA?

2/ Is YEATS2 required for growth of "normal" lung fibroblast cell lines such as the immortalized WI-38 cells?

3/ Fig. 5: Analysis of total histones in YEATS2 KD H1299 cells and A549 cells revealed a marked reduction in H3K9ac levels. What about H3K27ac?

H3K9ac ChIP-seq signals on the ZZZ3-occupied sites also showed reduction of H3K9ac in promoters (Fig. 5b). What about H3K27ac?

4/ Fig 5a: does KD of YEATS2 affect stability of ATAC complex components such as GCN5 and PCAF?

6/ Discussion section: The same YEATS domain was recently shown capable of binding to histone crotonylation more tightly (ref 28-30) but as authors correctly pointed out, cellular level of histone crotonylation is orders of magnitude lower than that of histone acetylation. Nevertheless, authors want to touch down in the discussion and include a possibility of binding to histone crotonylation by YEATS2 in the examined pathways.

Reviewer #2 (Remarks to the Author):

Mi et al. describe a novel reader protein for Histone 3 lysine 27 acetylation in "YEATS2 links histone acetylation to tumorigenesis on non-small cell lung cancer". This study extends previous findings that YEATS domains bind and "read" acetylated histones. They implicate this enzyme in lung cancer. However, the data suggest that YEATS2 may have more global roles in tumorigenesis. The study demonstrates that the reduction of YEATS2 causes reduced tumor cell and 3D growth (in vitro and in vivo). However, the authors do not formally prove the impact is only on cancer cells versus other cells. They also do not demonstrate whether there is a change in cell cycle or

apoptosis or whatever causes their impact on "proliferation". These are critical experiments in this study. Overall, the manuscript is interesting but has a number of points that need to be addressed so that the manuscript is stronger and more appropriate for Nature Communications.

Points to address:

1- Figure 1-The authors' mention that the TCGA data says that YEATS2 is amplified. The authors' need to clarify this data as whether focal amplification or whole arm events. For example, is YEATS2 actually amplified alone or other genes, which will determine whether the genetics supports a driver event or oncogenic amplification. How does the amplification correlate with expression. It is important to realize that many genes show amplification but this does not imply functional. Are other genes adjacent to YEATS2 amplified in the same samples and their expression increased?

They also focus on lung cancer but the data suggest other tumors too. It suggests more global impact on tumors. How does YEATS2 shRNA impact other cancer cell types. They need to establish whether the depletion of YEATS2 phenocopies the data in lung cancer cells because this would suggest they are following a more general process.

The authors MUST include data on what is happening to the cells. Are they exiting cell cycle? Are they undergoing apoptosis? Conduct FACS, etc.

The western for YEATS2 needs clarification on what is the real protein. There are multiple bands that suggest differential splice forms or modifications. These need to have clear marking for which match in Figure 1b and 1c. The authors' need to include a housekeeping protein in the western for 1b. They need to conduct quantification to show amount different in modification. The authors' need to control for HDAC activity if looking at acetylation and blot for multiple sites including K27acetyl.

Graphs throughout need to have the comparison to the "control" and the YEATS2 wild type. For example, 1e needs the comparison, which is likely not significant, between shNT+Vector and shY2+YEATS2.

The HR in 1f is not that impressive. Most studies require it to be closer to 2 to have meaning. This data should be related to 1a and in the supplement and should be cautiously mentioned. In fact, the authors' should evaluate in the other tumor types and see if trend is the same.

2- In Figure 2, the authors' use one strong shRNA. They need to compare data across shRNAs or optimally rescue and compare what really changes.

The authors' overlook for this paper that more genes are up-regulated. These data suggest that the protein has a repressive role. Why was this ignored? These genes should also be included in the analysis of Figure 4 and 5. After all, these targets may be drivers of the phenotypes. For example, lysosomes emerged as a category. Lysosome function is critical in cancer.

The authors' need to clarify the titles and statements that YEATS regulates K9acetylation at ribosomal genes- NOT globally. This is important for Figures 4 and 5 too. After all, Figure 5 only analyzes these ribosomal genes.

Need p values for each category Figure 1b.

3- Figure 3 combines peptide pulldowns and crystallography. These studies are nice, however, the complex substrates need to be included since peptide studies have provided artifacts in the literature for histones and nucleosomes. Pull downs should be included with these to highlight interaction. Do in vivo IPs show enrichment for the K27Acetyl?

If the authors' model other acetylated residues, do they model association. The peptide pulldown has a weak signal for H3K18acetyl. Does this have a major affinity different ITC? How does the affinity of YEATS2 compare to their recent studies related to AF9 YEATS binding?

Be careful saying a selective reader for K27acetylation since all in vitro data with peptides and nothing in vivo.

4- The authors' provide an explanation of why they study ZZZ3 but they need to conduct the studies +/- YEATS2 across genome not just ribosomal genes like in Figure 5. If the ZZZ3 protein is depleted, how does this impact K9acetylation and targets?

The data on K9acetylation is striking. Are KATs reduced in the YEATS2 shRNAs? If global K9acetylation drops, just promoters? Does this happen with the ATAC enzyme complex members too? If not, these data suggest more global impact and need to make sure studies are analyzed in that context and in relation to ATAC.

Are all K acetylations dropping? Need to blot for others. Are HDACs upregulated?

Figure 5 needs a new title since only evaluating ribosomal genes.

S. Figure 4A is nice but they need to see if global changes in H3 acetylation occur with ATAC member depletion and how this compares to the K9acetylation profiles.

Why did the authors' ignore K27acetylation patterns? Should evaluate too.

5- The authors' need to do stat comparisons as described above in Figure 6a,b. Stats are needed in comparing the shNT+Vector to the shY2+Vector.

6- Are tumors worse with overexpression in cancer or non-cancer cells? Afterall, this is the condition the authors' start with. In addition, they show the cancer cell lines have increased YEATS2- why? Amplified? Need to explain and discuss that other options impact expression.

Overall, the paper provides data consistent with YEATS2 reading K27acetylation. Whether this is the basis for the phenotypes is still not developed and proven. The authors' present data suggesting important for ATAC regulation of the ribosomal genes but ignore the fact that most genes up regulated with YEATS2 shRNA. The paper does not demonstrate that the phenotype is specific to lung cancer as the title implies and the paper does not show that the ribosomal genes are impacting the phenotype. The authors' should address these points above so the manuscript is stronger and appropriate for publication.

Reviewer #3 (Remarks to the Author):

Review paper Nature communications 2/1/2017 JHS
YEATS2 links histone acetylation to tumorigenesis of non-small cell lung cancer" by Professor Shi and colleagues

Research summary:

Though YEATS2 has been established as an acetyllysine reader and scaffolding protein of the ATAC complex, the question remained how it is recruited to a set of genes distinct from the SAGA target genes. Anticipating a mechanism where the subunits of ATAC direct this in a mechanism that

parallels the way sub-modules do this for SAGA, the authors aimed to characterize the molecular and biological functions of YEATS2 within the ATAC complex.

In this study, evidence is provided for a model where YEATS2 binds specifically H3K27Ac, delivering ATAC to its target genes to induce ATAC dependent promoter H3K9ac, favoring gene transcription. Notably, they elucidated this mechanism in a clinical relevant context, following the clue that YEATS2 is overexpressed in NSCLC tumorigenesis, leads to a hyperactive ATAC complex, and promoting expression of oncogenes.

In a set of coherent and conclusive studies the authors demonstrate that YEATS 2 is necessary for cancer growth and survival and required for expression of ribosome protein-encoding genes in a way that relies on YEATS H3K27ac recognition and ATAC-mediated H3K9 promoter acetylation.

Considerations; following the main text order

Figure 1a.

YEATS2 expression status from TCGA database:

The authors compare the gene expression levels across cancers, stressing the high levels for NSCLC. Later (line 147) the point is made that after YEATS2 knock-down, pathways are deregulated that are also deregulated in lung, colorectal and pancreatic cancers. It would be informative to see the gene expression levels of YEATS2 in those cancer types in Figure 1a.

Figure 1b.

This figure shows that not only YEATS2, but also PCAF and GCN5 are upregulated in lung cancer cell lines. Thus, the argument is that having a surplus of the essential subunits of ATAC might lead to a hyperactive complex, leading to increased acetylation on H3K9 and H3K14. The authors state that especially H3K9ac was evidently higher in the NSCLC lines. These blots are however not convincing. To strengthen this claim, a quantification with replicates should be presented, demonstrating that H3K9 and/or H3K14 are truly increased in the NSCLC compared to the immortal cell lines.

Supplemental figure 1.d

This figure demonstrates that YEATS2 contributes to lung cancer cell growth/survival in lines H1299 and A549. Is a similar dependency on YEATS 2 observed in a line without upregulated GCN5 and/or PCAF, like H520? It would be interesting if a lower abundance of these essential subunits enhances the effect of YEATS2 knock down on growth/survival.

Figure 2b

To get insights into how YEATS2 mediates tumorigenesis, RNA-seq was performed in WT and YEATS2-shRNA. (Please clarify what cell line was used for these experiments).

Specific pathways are found to be up and downregulated after knockdown. Next, it is explained that since YEATS2/ATAC is mostly involved in gene activation, the paper will focus on genes downregulated after YEATS2 depletion. Though this is a fair argument, and one can only follow up on so many findings, it is intriguing to reverse the argument and ask the question that would hint more to the biological mechanism as to how YEATS2 promotes tumorigenesis of NSCLC. In the NSCLC, what genes are downregulated after overexpression of YEATS2; i.e. what genes are derepressed after the knockdown. From the description it seems that these are involved in glucan degradation and focal adhesion, intuitively mechanisms that promote metastasis, which would conflict with a role for YEATS2 in promoting tumorigenesis and the increased growth/survival as had been demonstrated after YEATS2 shRNA. Rather than asking for additional experiments, we'd like to hear the author's speculations on these observations.

Figure 3c

Next, it is appropriately tested what substrate YEATS2 binds, and it is reported to be H3K27ac with a Kd of 274uM. How does this relate to the other reported affinities for YEATS-acetyl binding? i.e.,

could you provide some context for the readership less familiar with these values? From later descriptions on the co-crystal structure, it seems to be rather weak interaction.

Lines 199-208

In this paragraph it is described how residues surrounding K27 are important for the binding to YEATS2. Mutations in the sandwich pocket disturbed the binding. Reciprocally, do substitutions in the histone peptide prevent stable interactions? i.e., does this argue for a requirement of a specific set of histone modifications around K27 for a stable binding in the WT sandwich pocket?

Section ChIP experiments (line 210 & Cn'd)

Since YEATS2 wasn't ChIP-able, the authors found a work around via chipping ZZZ3, an ATAC-specific subunit in order to study if YEATS2 co-localizes with H3K27ac and H3K9ac, and reported colocalization of ZZZ3 with these marks and predominantly at promoter regions.

Though this study has a clear focus on acetylation, in the light of recent discoveries we cannot ignore that TAC might be targeted via other acylations to its' target sites. Though these are less abundant as argued in the discussion, they might reflect an important mechanism given their link to metabolism, and the fact that metabolic changes are a hallmark of cancer. Especially during disease states, these modifications might be more abundant and reflect a gene-regulatory response to the diseased state, either inductive of subdue.

Thus, how does ATAC co-localize to a pan-KCr chip in NSCLC, a site-specific Kcr ChIP, or even to data in WT from literature like the k318Kcr data from Sabri et. al 2015?

Figure 5a

How do the authors account for the lack of reduction in H3K14ac after YEATS2 KD?

Responses to Reviewers

We thank all of our reviewers for their insightful and positive comments. On the basis of the collective reviewers' comments, we have revised the manuscript and included several sets of new data to provide greater insights into the molecular mechanism and biological functions of the recognition of histone H3 acetylation by the YEATS2 YEATS domain in non-small cell lung cancer. The results of the new experiments, in conjunction with our previous findings, demonstrate that YEATS2 is a histone H3K27 acetylation reader that regulates a transcriptional program essential for cancer cell growth and survival.

Major additions include:

- 1) Cell proliferation assays in a panel of NSCLC cells and ovarian cancer cells that harbor YEATS2 amplification, as well as in immortalized "normal" lung fibroblast cell lines. The results suggest that YEATS2 is an essential gene in a broad range of cancer cell lines as well as "normal" cells.
- 2) Analysis of acetylation on histone H3K27 in addition to H3K9ac. The results suggest that depletion of YEATS2 also affects gene-specific and global H3K27ac levels.
- 3) Analysis of upregulated genes in YEATS2 knockdown cells. Combination of ChIP-seq and RNA-seq analyses and ChIP-qPCR data in YEATS2 and ZZZ3 KD cells suggest that the upregulated genes are likely not direct targets of the ATAC complex.
- 4) Functional experiments to determine consequence of transcriptional dysregulation upon YEATS2 knockdown. The results suggest that YEATS2 depletion inhibits cell proliferation by cell cycle arrest but not apoptosis.

In summary, we believe that these and other additional results further strengthen our conclusions and enhance the overall significance of the manuscript. Below are our point-by-point responses to the reviewers' comments.

Reviewers' comments:

Reviewer #1 (Remarks to the Author):

This work of Mi et al reports that YEATS2, a gene encoding a human YEATS domain-containing chromatin reader, is highly amplified in cancer including non-small cell lung cancer (NSCLC). Authors found YEATS2 is required for NSCLC growth and transforming potential, partly through maintaining essential gene expression programs such as ribosomal proteins to sustain cell proliferation. The YEATS domain of YEATS2 is a selective reader for H3K27ac. ChIP-seq of a YEATS2-associated complex component also reveals its binding mainly at gene promoters with H3K27ac. Lastly, authors carried out point mutagenesis at the YEATS domain to demonstrate that reading of H3K27ac is essential for recruitment of YEATS2-associated HAT complex

(ATAC) to promoter, ATAC-mediated histone acetylation (H3K9ac), and cancer-promoting functions.

This work is done by the team who initially made discovery of YEATS domains as a novel “reader” class specific to histone acetylation. A novel finding of this current work is to link YEATS2 and its histone H3K27ac-selective “reader” function to cancer. This study uses an integrated approach involving cancer cell biology, structural biology and genomics profiling to delineate mechanistic details of a cancer related pathway, which was not reported before. The implication of the work is far reaching in terms of therapeutics. For example, targeting the histone acetylation reader domain of BRD4 is now widely accepted as a promising anti-cancer strategy. Therefore, the report is also timely and shall appeal to the field and readers.

Comments:

1/ Fig. 1f: author needs to state the source of data for prognosis of human lung cancer patients. Is it from TCGA?

The data is obtained from the Kaplan Meier plotter server (<http://kmplot.com>). This server is capable to assess the effect of 54,675 genes on patient survival using 10,461 cancer samples, including 2,437 NSCLC patients with a mean follow-up of 49 months (Szasz et al., 2016). We have now included this information in the figure legend.

2/ Is YEATS2 required for growth of “normal” lung fibroblast cell lines such as the immortalized WI-38 cells?

This is a great point! We KD YEATS2 in immortalized “normal” lung fibroblast cell lines WI-38 and IMR-90 and we observed inhibition of proliferation in both cell lines, suggesting that YEATS2 is an essential gene not only for cancer cells, but also for normal cells. This data is now included in Supplementary Fig. 2h,i.

Supplementary Figure 2h,i. YEATS2 is required for the growth of immortalized normal lung fibroblast cells. Cell proliferation assay of WI38 (h) or IMR90 cells (i) treated with control (shNT) or YEATS2 shRNAs (shY2). Right panels: qRT-PCR analysis showing YEATS2 shRNA knockdown efficiency.

3/ Fig. 5: Analysis of total histones in YEATS2 KD H1299 cells and A549 cells revealed a marked reduction in H3K9ac levels. What about H3K27ac? H3K9ac ChIP-seq signals on the ZZZ3-occupied sites also showed reduction of H3K9ac in promoters (Fig. 5b). What about H3K27ac?

This is also a great point! By Western blot analysis, we observed a reduction in global H3K27ac levels upon YEATS2 KD (Fig. 5a). We also performed H3K27ac ChIP-seq, and observed reduced H3K27ac levels on ZZZ3-occupied genes in YEATS2 KD cells (Supplementary Fig. 5c). This reduction might be an indirect effect as the catalytic subunit of ATAC complex, PCAF (p300/CBP associated factor), is known to directly interact with p300/CBP, the dominant HATs for H3K27ac in cells (Marmorstein, 2001).

Knockdown of YEATS2 reduces H3K27ac levels.

Left panel: Western blot analysis of histone acetylation in control and YEATS2 KD cells (Fig. 5a.)
 Right panel: Average H3K27ac occupancy on the promoter of the ZZZ3 bound genes or unbound genes (others) in control (shNT) and YEATS2 KD (shY2) H1299 cells (Supplementary Fig. 5c).

4/ Fig 5a: does KD of YEATS2 affect stability of ATAC complex components such as GCN5 and PCAF?

YEATS2 KD has no or little affect on stability of the ATAC complex components. These data are now included in Supplementary Fig. 5b.

Supplementary Figure 5b. Western blot analysis of the ATAC complex components and HDAC1 in control (shNT) and YEATS2 KD (shY2) cells. Actin is shown as a loading control.

6/ Discussion section: The same YEATS domain was recently shown capable of binding to histone crotonylation more tightly (ref 28-30) but as authors correctly pointed out, cellular level of histone crotonylation is orders of magnitude lower than that of histone acetylation. Nevertheless, authors want to touch down in the discussion and include a possibility of binding to histone crotonylation by YEATS2 in the examined pathways.

We thank the reviewer for the great suggestion. We've attempted to ChIP for H3K27cr in H1299 cells, but failed to detect H3K27cr by sequencing. This may be due to the low abundance of this modification, or the lack of ChIP-grade antibody. Nevertheless, as suggested, we've discussed this important point in the Discussion section.

Reviewer #2 (Remarks to the Author):

Mi et al. describe a novel reader protein for Histone 3 lysine 27 acetylation in “YEATS2 links histone acetylation to tumorigenesis on non-small cell lung cancer”. This study extends previous findings that YEATS domains bind and “read” acetylated histones. They implicate this enzyme in lung cancer. However, the data suggest that YEATS2 may have more global roles in tumorigenesis. The study demonstrates that the reduction of YEATS2 causes reduced tumor cell and 3D growth (in vitro and in vivo). However, the authors’ do not formally prove the impact is only on cancer cells versus other cells. They also do not demonstrate whether there is a change in cell cycle or apoptosis or whatever causes their impact on “proliferation”. These are critical experiments in this study. Overall, the manuscript is interesting but has a number of points that need to be addressed so that the manuscript is stronger and more appropriate for Nature Communications.

Points to address:

1- Figure 1-The authors’ mention that the TCGA data says that YEATS2 is amplified. The authors’ need to clarify this data as whether focal amplification or whole arm events. For example, is YEATS2 actually amplified alone or other genes, which will determine whether the genetics supports a diver event or oncogenic amplification. How does the amplification correlate with expression. It is important to realize that many genes show amplification but this does not imply functional. Are other genes adjacent to YEATS2 amplified in the same samples and their expression increased?

This is a great point! YEATS2 is located within the chromosomal segment 3q26, a region frequently amplified in lung squamous cell carcinoma (LSCC, Supplementary Fig. 1a) and many other types of cancers (Fields et al., 2016; Fong and Minna, 2002; Larsen and Minna, 2011; Qian and Massion, 2008; Qian et al., 2015). The size of the amplicon varies between tumors, spanning from chromosome 3q22 to 3qter with a most frequent region of amplification in squamous cell carcinoma being between 3q26 and 3q28 (Qian et al., 2015). Previous studies have revealed that among the amplified 3q genes, 12 genes, including YEATS2, are significantly correlated in LSCC (Qian et al., 2015).

Analysis of the TCGA data reveals that amplification of YEATS2 and its neighboring genes KLHL24 and MAP6D1 correlates with gene expression levels very well in LSCC (Fig. R1). Despite the high frequency of amplification and overexpression in human cancers, comparing to some other amplified genes, such as PIK3CA that have been extensively studied (Sequist et al., 2011; Umemura et al., 2015), the role of YEATS2 in oncogenesis remains largely unknown. In this study, collectively we show that YEATS2 is overexpressed in NSCLC and plays an important role in maintaining histone acetylation and expression of genes essential for cancer cell growth and survival.

Amplification of YEATS2 and neighboring genes correlates with elevated gene expression in lung squamous cell carcinoma.

Left panel: Integrated analysis of YEATS2 amplification in 1020 LSCC samples. Frequency plots of the copy-number abnormalities indicate degree of copy-number loss (blue) or gain (red). The color intensity indicates the extent of copy-number changes. Representative genes in the amplicons of chromosome 3q27.1 and 3126.32 are shown (Supplementary Fig. 1a) Right panel: mRNA levels of YEATS2 and its neighboring genes (KLHL24 and MAP6D1) are positively correlated with gene amplification status in LSCC. PIK3CA gene is also shown for comparison. Data were obtained from TCGA (Fig. R1).

They also focus on lung cancer but the data suggest other tumors too. It suggest more global impact on tumors. How does YEATS2 shRNA impact other cancer cell types. They need to establish whether the depletion of YEATS2 phenocopies the data in lung cancer cells because this would suggest they are following a more general process.

This is also a great point. The TCGA data show that in addition to LSCC, YEATS2 is also amplified and overexpressed in many other types of cancers including ovarian cancer (Supplementary Fig. 1c). We KD YEATS2 in two ovarian cell lines, CaoV3 and HeyA8, that also harbor YEATS2 amplification, and we observed inhibition of proliferation by YEATS2 KD (Supplementary Fig. 2f,g). Together, these data suggest that YEATS2 is an essential gene required not only for lung cancer cells, but also other types of cancer cells harboring YEATS2 amplification.

YEATS2 is amplified in ovarian cancer and is required for ovarian cancer cell growth.

Left panel: YEATS2 mRNA levels are positively correlated with gene amplification status in ovarian cancer (Supplementary Fig. 1c).

Middle and right panels: Cell proliferation assay of ovarian cancer cells CaoV3 (f) and HeyA8 (g) treated with control (shNT) or YEATS2 (shY2) shRNAs. Western blots show YEATS2 shRNA knockdown efficiency (Supplementary Fig. 2f,g).

The authors MUST include data on what is happening to the cells. Are they existing cell cycle? Are they undergoing apoptosis? Conduct FACS, etc.

We thank the reviewer for the suggestions. We determined cell cycle and apoptosis in the YEATS2 KD cells. The data revealed that YEATS2 KD led to G1 arrest of cell cycle but had little effect on apoptosis. These data are now included in Supplementary Fig. 3c,d.

Supplementary Figure 3c,d. YEATS2 KD affects cell cycle progression but not apoptosis. Flow cytometry cell cycle analysis (c) and apoptosis analysis (d) of control (shNT) and YEATS2 knockdown (shY2-1) H1299 cells.

The western for YEATS2 needs clarification on what is the real protein. There are multiple bands that suggest differential splice forms or modifications. These need to have clear marking for which match in Figure 1b and 1c. The authors' need to include a housekeeping protein in the western for 1b. They need to conduct quantification to show amount different in modification. The authors' need to control for HDAC activity if looking at acetylation and blot for multiple sites including K27acetyl.

We thank the reviewer for the suggestions. The YEATS2 antibody shows cross reactivity to some non-specific bands. In the figures, we used arrow to mark the endogenous YEATS2 proteins (based on KD results). As suggested, we also included actin, HDAC1, H3K27ac, as well as quantifications of histone acetylation blots in Fig. 1b.

Figure 1b. Western blot analysis of YEATS2, GCN5, PCAF, HDAC1, H3K9ac and H3K14ac in indicated NSCLC cell lines and immortalized “normal” lung fibroblast cell lines. Total H3 and actin are shown as a loading control. The arrow indicates the band of YEATS2 protein. Quantifications of H3K9ac, H3K14ac and H3K27ac are shown in the right panel.

Graphs throughout need to have the comparison to the “control” and the YEATS2 wild type. For example, 1e needs to be compared, which is likely not significant, between shNT+Vector and shY2+YEATS2.

We have now included comparison between control and the YEATS2 wild type in all bar graphs.

The HR in 1f is not that impressive. Most studies require it to be closer to 2 to have meaning. This data should be related to 1a and in the supplement and should be cautiously mentioned. In fact, the authors’ should evaluate in the other tumor types and see if trend is the same.

We thank the reviewer for the suggestions. The data is obtained from the webserver: <http://kmplot.com>. The Kaplan Meier plotter server is currently capable to assess gene expression in breast, ovarian, lung and gastric cancer patients (Szasz et al., 2016). As ovarian cancer patients also harbor frequent YEATS2 amplification (28%), we also determined the correlation of YEATS2 expression with progression-free survival in ovarian cancer patients. The results show that as in NSCLC, high YEATS2 expression levels are correlated with worse prognosis of ovarian cancer patients. As suggested, we have now moved Fig. 1f into Supplementary Fig. 1e, and included the ovarian cancer data in Supplementary Fig. 1f.

Supplementary figure 1e,f. High YEATS2 expression is correlated with worse prognosis. Kaplan-Meier survival curves of NSCLC (e) or ovarian cancer patients (f) with low or high YEATS2 expression levels. Data were obtained from Kaplan Meier plotter (<http://kmplot.com/>).

2- In Figure 2, the authors’ use one strong shRNA. They need to compare data across shRNAs or optimally rescue and compare what really changes.

As suggested, we have now performed RNA-seq analysis using a second shRNA (shY2-2). ShY2-2 does not KD YEATS2 as efficiently as shY2-1 (Fig. 1c). Nevertheless we identified 520 overlapped genes downregulated in both YEATS2 KD cells, which include 29 genes involved in ribosomal RNA processing and 11 ribosomal protein (RP) genes (Supplementary Fig. 3a,b). Furthermore, we have also shown in other figures that downregulation of RP genes can be validated by independent quantitative real-time RT-PCR using both shRNAs (Fig. 2d), and can be rescued by WT YEATS2 (Fig. 6b). The new RNA-seq data is now included in Supplementary Fig. 3a,b and Supplementary Table 1.

Supplementary figure 3a. Heatmap representation of differentially expressed genes in control (shNT) and YEATS2 knockdown (shY2-2) cells from two independent biological replicates.
Supplementary figure 3b. Overlaps of down-regulated genes in cells treated with two independent shRNAs. Overlapped down-regulated RP genes are listed.

The authors' overlook for this paper that more genes are Up-regulated. These data suggest that the protein has a repressive role. Why was this ignored? These genes should also be included in the analysis of Figure 4 and 5. After all, these targets maybe drive of the phenotypes. For example, lysosome emerged as a category. Lysosome function is critical in cancer.

This is an important point. We thank the reviewer for the suggestions. RNA-seq analysis identified 1,748 genes downregulated whereas 3,361 genes upregulated. However, comparison of RNA-seq and ZZZ3 ChIP-seq data suggests only small number genes are direct targets of the YEATS2/ATAC complex (Supplementary Fig. 5d). KEGG pathway analysis reveals that among the down-regulated direct target genes, 39 are enriched in the pathway of ribosome and 12 in DNA replication, whereas among upregulated genes, only 10 genes enriched in two pathways (Supplementary Fig. 5e,f). Importantly, KD of YEATS2 markedly reduced ZZZ3 occupancy and H3K9ac levels on the downregulated RP genes (Fig. 5c-e), but had little or no effect on the 10 up-regulated genes (Supplementary Fig. 5g, h). Furthermore, KD of ZZZ3 did not affect the ChIP signals observed on these genes (Fig. R2). Together, all these data suggest that these upregulated genes are likely not direct target of YEATS2 or the ATAC complex.

Supplementary figure 5d-h. YEATS2 KD does not affect H3K9ac levels and ZZZ3 occupancy on the upregulated genes in YEATS2 KD cells. (d) Venn diagram showing the overlap of ZZZ3 occupied genes with downregulated or upregulated genes in YEATS2 KD cells. (e,f) KEGG pathway analysis of down- (e) or up-regulated (f) ZZZ3 occupied genes. (g,h) qPCR analysis of H3K9ac (g) or ZZZ3 ChIP (h) in control (shNT) and YEATS2 KD (shY2) H1299 cells of the 10 upregulated genes as in (f). **Fig. R2:** qPCR analysis of ZZZ3 ChIP in control (shNT) and ZZZ3 KD (shZZZ3) cells.

The authors' need to clarify the titles and statements that YEATS regulates K9acetylation at ribosomal genes- NOT globally. This is important for Figures 4 and 5 too. After all, Figure 5 only analyzes these ribosomal genes.

We thank the reviewer for the suggestion. We have changed the titles of Fig. 4 and Fig. 5 to specify the changes at ribosomal protein encoding genes. It is worth noting that the reduction in H3K9ac levels can be seen on all ZZZ3 bound genes (Fig. 5b).

Need p values for each category Figure 2b.

The X-axis denotes p value for each category.

3- Figure 3 combines peptide pulldowns and crystallography. These studies are nice, however, the complex substrates need to be included since peptide studies have provided artifacts in the literature for histones and nucleosomes. Pull downs should be included with these to highlight interaction. Do in vivo IPs show enrichment for the K27Acetyl?

We've now included data in Fig. 3c and 3d showing the interaction between YEATS2 and H3K27ac at full-length histone and nucleosomal levels by *in vitro* histone binding and *in vivo* IP, respectively.

Figure 3c,d. YEATS2 binds to H3K27ac in FL histone *in vitro* and nucleosomes in cells.

If the authors' model other acetylated residues, do they model association. The peptide pulldown has a weak signal for H3K18acetyl. Does this have a major affinity different ITC? How does the affinity of YEATS2 compare to their recent studies related to AF9 YEATS binding?

We've now compared YEAYS2 binding to acetylation on different lysines of H3 and the un-modified H3 tail side-by-side. ITC titrations show that the YEATS2 YEATS domain has the strongest binding to H3K27ac ($K_D=50\mu\text{M}$), modest binding to H3K18ac ($K_D=120\mu\text{M}$), whereas no detectable bindings to H3K9ac and H3K14ac (Fig. 3e).

Previously we've detected the binding affinity of the AF9 YEATS domain to H3K27ac $\sim 7\mu\text{M}$ (Li et al., 2014), indicating a higher affinity to H3K27ac than that of YEATS2 ($K_D=50\mu\text{M}$). Nevertheless, AF9 YEATS shows an even higher affinity to H3K9ac ($K_D=3.7\mu\text{M}$), whereas YEATS2 has no detectable binding to H3K9ac. These results suggest that different YEATS domains have distinct substrate preferences, wherein YEATS2 prefers H3K27ac over other acetylated H3 peptides.

Figure 3e. ITC titration curves of the YEATS2 YEATS domain with histone peptides.

Be careful saying a selective reader for K27acetylation since all in vitro data with peptides and nothing in vivo.

We thank the reviewer for the suggestion and we have modified our statement accordingly.

4- The authors' provide an explanation of why they study ZZZ3 but they need to conduct the studies +/- YEATS2 across genome not just ribosomal genes like in Figure 5. If the ZZZ3 protein is depleted, how does this impact K9acetylation and targets?

This is a great suggestion. We've carried out ChIP-seq analysis of both ZZZ3 and H3K9ac in control and YEATS2 KD cells to determine changes in their occupancies on ZZZ3 target genes and across the genome. YEATS2 KD reduced H3K9ac and ZZZ3 occupancies on ZZZ3 bound genes (Fig. 5b, 5f). In contrast, the H3K9ac levels on non-ZZZ3 bound genes (denoted as others) were largely not affected (Fig. 5b). Similarly, KD of ZZZ3 also reduced H3K9ac levels on its target genes (Fig. R3).

Depletion of YEATS2 or ZZZ3 reduces H3K9ac levels on ZZZ3/ATAC target genes.

Left and middle panels: Average occupancy of H3K9ac (Fig. 5b) or ZZZ3 (Fig. 5f) on the promoter of the ZZZ3-occupied genes in control (shNT, blue) and YEATS2 KD (shY2, red) in H1299 cells. non-ZZZ3 bound genes (others) are shown as black and gray lines.
 Right panel: Average occupancy of H3K9ac on the promoter of the ZZZ3-occupied genes in control (shNT, blue) and ZZZ3 KD (shZZZ3, red) in H1299 cells (Fig. R3).

The data on K9acetylation is striking. Are KATs reduced in the YEATS2 shRNAs? If global K9acetylation drops, just promoters? Does this happen with the ATAC enzyme complex members too? If not, these data suggest more global impact and need to make sure studies are analyzed in that context and in relation to ATAC.

YEATS2 KD has no or little affect on the protein stability of GCN5/PCAF or other ATAC complex components (Supplementary Fig. 5b). As H3K9ac peaks are mainly localized in promoter regions, the reduction of H3K9ac on ZZZ3 target genes is not surprisingly at promoters. The reduction of global H3K9ac levels is dependent on the ATAC complex, as depletion of ZZZ3 or GCN5 reduces global H3K9ac levels (Fig. R4).

Fig. R4. Depletion of the ATAC complex component reduces global H3K9ac levels in cells. Western blot analysis of H3K9ac levels in control (shNT) and GCN5 or ZZZ3 KD cells.

Are all K acetylations dropping? Need to blot for others. Are HDACs upregulated?

This is a great point. We've examined acetylation on several histone residues, and we found that YEATS2 KD greatly reduced H3K9ac, and, to lesser extends, the levels of H3K27ac, H3K14ac, H4K16ac and H4-tetra acetylation. However, we did not observed HDAC1 upregulated. These data are now included in Fig. 5a and Supplementary Fig. 5b.

Knockdown of YEATS2 reduces global histone H3 acetylation levels.

Left panel: Western blot analysis of histone acetylation in control and YEATS2 KD cells (Fig. 5a.)
 Right panel: Western blot analysis of the ATAC complex components and HDAC in control (shNT) and YEATS2 KD (shY2) cells (Supplementary Fig. 5b).

Figure 5 needs a new title since only evaluating ribosomal genes.

S. Figure 4A is nice but they need to see if global changes in H3 acetylation occur with ATAC member depletion and how this compares to the K9acetylation profiles.

We thank the reviewer for the suggestions. We have changed the title of Fig. 5 to specify the changes at ribosomal protein encoding genes. As shown in Figure R4, the reduction of global H3K9ac is dependent on ATAC complex, as depletion of ZZZ3 or GCN5 also leads to reduction of global H3K9ac levels.

Why did the authors' ignore K27acetylation patterns? Should evaluate too.

This is a great point. We determined global and gene-specific H2K27ac levels by Western blot and ChIP-seq analyses. YEATS2 KD reduced global H3K27ac levels as well as H3K27ac on ZZZ3-occupied genes. These data are included in Fig. 5a and Supplementary Fig. 5c.

Supplementary Figure 5c. Knockdown of YEATS2 reduces H3K27ac levels.

Average H3K27ac occupancy on the promoter of the ZZZ3 bound genes or unbound genes (others) in control (shNT) and YEATS2 KD (shY2) H1299 cells.

5- The authors' need to do stat comparisons as described above in Figure 6a,b. Stats are needed in comparing the shNT+Vector to the shY2+Vector.

We have now included statistic comparison between shNT+Vector and shY2+Vector.

6- Are tumors worse with overexpression in cancer or non-cancer cells? Afterall, this is the condition the authors' start with. In addition, they show the cancer cell lines have increased YEATS2- why? Amplified? Need to explain and discuss that other options impact expression.

YEATS2 is located within the chromosomal segment 3q26, a region frequently amplified in NSCLC and certain other types of cancers such as ovarian cancer (Fields et al., 2016; Fong and Minna, 2002; Larsen and Minna, 2011; Qian and Massion, 2008; Qian et al., 2015). Amplification is considered as the main mechanism of YEATS2 overexpression in cancers. YEATS2 amplification status is positively correlated to its gene expression levels in NSCLC and ovarian cancers (Supplementary Fig. 1b,c), and high expression levels of YEATS2 are associated with worse prognosis of NSCLC and ovarian cancer patients (Supplementary Fig. 1e,f). We thank the reviewer for raising these important points and we have carefully discussed this in the paper.

Overall, the paper provides data consistent with YEATS2 reading K27acetylation. Whether this is the basis for the phenotypes is still not developed and proven. The authors' present data suggesting important for ATAC regulation of the ribosomal genes but ignore the fact that most genes up regulated with YEATS2 shRNA. The paper does not demonstrate that the phenotype is specific to lung cancer as the title implies and the paper does not show that the ribosomal genes are impacting the phenotype. The authors' should address these points above so the manuscript is stronger and appropriate for publication.

We thank this reviewer for all the critical questions, which we have addressed in our point-by-point responses. On the basis of the comments, the results of the new experiments have greatly improved the quality of this paper.

Reviewer #3 (Remarks to the Author):

*Review paper Nature communications 2/1/2017 JHS
YEATS2 links histone acetylation to tumorigenesis of non-small cell lung cancer" by
Professor Shi and colleagues*

Research summary:

Though YEATS2 has been established as an acetyllysine reader and scaffolding protein of the ATAC complex, the question remained how it is recruited to a set of genes distinct from the SAGA target genes. Anticipating a mechanism where the subunits of ATAC direct this in a mechanism that parallels the way sub-modules do this for SAGA, the authors aimed to characterize the molecular and biological functions of YEATS2 within the ATAC complex.

In this study, evidence is provided for a model where YEATS2 binds specifically H3K27Ac, delivering ATAC to its target genes to induce ATAC dependent promoter H3K9ac, favoring gene transcription. Notably, they elucidated this mechanism in a clinical relevant context, following the clue that YEATS2 is overexpressed in NSCLC tumorigenesis, leads to a hyperactive ATAC complex, and promoting expression of oncogenes.

In a set of coherent and conclusive studies the authors demonstrate that YEATS 2 is necessary for cancer growth and survival and required for expression of ribosome protein-encoding genes in a way that relies on YEATS H3K27ac recognition and ATAC-mediated H3K9 promotor acetylation.

Considerations; following the main text order

Figure 1a.

YEATS2 expression status from TCGA database:

The authors compare the gene expression levels across cancers, stressing the high levels for NSCLC. Later (line 147) the point is made that after YEATS2 knock-down, pathways are deregulated that are also deregulated in lung, colorectal and pancreatic cancers. It would be informative to see the gene expression levels of YEATS2 in those cancer types in Figure 1a.

We thank the reviewer for the suggestion. We have now included the YEATS2 gene expression data in Supplementary Fig. 1d.

Supplementary Figure 1d. YEATS2 mRNA levels across different cancer types (TCGA).

Figure 1b.

This figure shows that not only YEATS2, but also PCAF and GCN5 are upregulated in lung cancer cell lines. Thus, the argument is that having a surplus of the essential subunits of ATAC might lead to a hyperactive complex, leading to increased acetylation on H3K9 and H3K14. The authors state that especially H3K9ac was evidently higher in the NSCLC lines. These blots are however not convincing. To strengthen this claim, a quantification with replicates should be presented, demonstrating that H3K9 and/or H3K14 are truly increased in the NSCLC compared to the immortal cell lines.

We thank the reviewer for the suggestion. We have now included quantification of the blots in the figure.

Supplemental figure 1.d

This figure demonstrates that YEATS2 contributes to lung cancer cell growth/survival in lines H1299 and A549. Is a similar dependency on YEATS 2 observed in a line without upregulated GCN5 and/or PCAF, like H520? It would be interesting if a lower abundance of these essential subunits enhances the effect of YEATS2 knock down on growth/survival.

This is a great point! We KD YEATS2 in H520 and Ludlu-1 cells and we also observed growth inhibition. The defect in cell proliferation seems more severe than in H1299 and A549 cells, indicating that lower abundance of essential ATAC subunit such as PCAF may enhance the effect of YEATS2 knockdown on cell growth. This data is now included in Supplementary Fig. 2d,e.

Supplementary Figure 2d,e. Cell proliferation assay of NSCLC cells treated with control (shNT) or YEATS2 (shY2) shRNAs. Western blots show YEATS2 shRNA knockdown efficiency.

Figure 2b

To get insights into how YEATS2 mediates tumorigenesis, RNA-seq was performed in WT and YEATS2-shRNA. (Please clarify what cell line was used for these experiments).

Specific pathways are found to be up and downregulated after knockdown. Next, it is explained that since YEATS2/ATAC is mostly involved in gene activation, the paper will focus on genes downregulated after YEATS2 depletion. Though this is a fair argument, and one can only follow up on so many findings, it is intriguing to reverse the argument and ask the question that would hint more to the biological mechanism as to how YEATS2 promotes tumorigenesis of NSCLC. In the NSCLC, what genes are downregulated after overexpression of YEATS2; i.e. what genes are derepressed after the knockdown. From the description it seems that these are involved in glucan degradation and focal adhesion, intuitively mechanisms that promote metastasis, which would conflict with a role for YEATS2 in promoting tumorigenesis and the increased growth/survival as had been demonstrated after YEATS2 shRNA. Rather than asking for additional experiments, we'd like to hear the author's speculations on these observations.

We thank the reviewer for raising this great point. RNA-seq analysis assesses global gene expression, identifying both direct and indirect genes dysregulated upon perturbation. In this study, upon YEATS2 depletion, downregulated genes are enriched in the pathways regulating ribosome biogenesis, DNA replication, cell cycle, DNA repair, and splicing, and upregulated genes enriched in lysosome functions, glycan degradation and focal adhesion (Fig. 2b). As KEGG pathway analysis does not distinguish positive regulators from negative regulators of pathway, upregulation of gene expression in a pathway does not always correlate with elevated pathway activity. For example, although we identified upregulated genes enriched in some pathways that promote metastasis, we did not observe increased migration activity in YEATS2 KD cells (Supplementary Fig. 3e). Furthermore, as we are interested in understanding whether the YEATS domain is important for the function of YEATS2 in chromatin and transcriptional regulation, in this

study we focus on direct target genes of YEATS2. Several lines of evidence collectively demonstrate that the downregulated RP genes are direct target genes of YEATS2 (Fig. 5c-e), whereas the upregulated genes are likely not (Supplementary Fig. 5g,h).

Supplementary Figure 3e. YEATS2 does not regulate cell migration. Representative views and quantification of trans-well cell migration assay in control (shNT) and YEATS2 (shY2) cells.

Figure 3c

Next, it is appropriately tested what substrate YEATS2 binds, and it is reported to be H3K27ac with a K_D of 274 μ M. How does this relate to the other reported affinities for YEATS-acetyl binding? i.e., could you provide some context for the readership less familiar with these values? From later descriptions on the co-crystal structure, it seems to be rather weak interaction.

During the revision we optimized the ITC titration conditions and we were able to measure a higher binding affinity of YEATS2 YEATS with the H3K27ac peptide ($K_D = 50\mu\text{M}$) (Fig. 3e). Nevertheless, the binding affinity is weaker than that of the AF9 or ENL YEATS domain, which shows a K_D value of 7 μM and 30 μM , respectively, to the H3K27ac peptide (Li et al., 2014; Wan et al., 2017). However, binding affinity *in vitro* does not always correlate with functional importance in cells. For instance, although *in vitro* ENL binding to histone acetylation is weaker than AF9, in MLL-rearranged leukemias, ENL, but not AF9, is essential for AML maintenance (Wan et al., 2017). In the current study, our results collectively demonstrate that the recognition of H3K27ac by the YEATS domain is important for the functionality of YEATS2 in NSCLC.

Lines 199-208

In this paragraph it is described how residues surrounding K27 are important for the binding to YEATS2. Mutations in the sandwich pocket disturbed the binding. Reciprocally, do substitutions in the histone peptide prevent stable interactions? i.e., does this argue for a requirement of a specific set of histone modifications around K27 for a stable binding in the WT sandwich pocket?

This is also a great point. We mutated two residues of H3, S28 and P30, to alanine and performed ITC titrations to test the importance of residues around H3K27. S28A did not affect YEATS2 binding to H3K27ac dramatically whereas P30A mutation decreased the binding about two-fold. These results indicate that the sequence flanking K27 is required, but likely not essential, for the stable binding around the YEATS2 YEATS sandwich pocket. These data is now included in Supplementary Fig. 4e.

Supplementary Figure 4c. ITC titration fitting curves of the YEATS2 YEATS domain with the indicated histone peptides.

Section ChIP experiments (line 210 & Cn'd)

Since YEATS2 wasn't ChIP-able, the authors found a work around via chipping ZZZ3, an ATAC-specific subunit in order to study if YEATS2 co-localizes with H3K27ac and H3K9ac, and reported colocalization of ZZZ3 with these marks and predominantly at promoter regions.

Though this study has a clear focus on acetylation, in the light of recent discoveries we cannot ignore that ATAC might be targeted via other acylations to its' target sites. Though these are less abundant as argued in the discussion, they might reflect an important mechanism given their link to metabolism, and the fact that metabolic changes are a hallmark of cancer. Especially during disease states, these modifications might be more abundant and reflect a gene-regulatory response to the diseased state, either inductive of subdue.

Thus, how does ATAC co-localize to a pan-KCr chip in NSCLC, a site-specific Kcr ChIP, or even to data in WT from literature like the k318Kcr data from Sabri et. al 2015?

We thank the reviewer for raising this great point. We agree with the reviewer that although at low abundance, histone crotonylation might reflect an important mechanism for stress response or in certain diseases. Thus it is worth looking into the relationship between YEATS2 and histone crotonylation. In this regard, we attempted to ChIP for H3K27cr in H1299 cells but failed to detect H3K27cr occupancy by sequencing. This is likely due to a low abundance of this modification, or the lack of ChIP-grade antibody. Nevertheless, in our previous effort, we've determined H3K9cr distribution in HeLa cells by ChIP-seq, and the data suggest that under normal growth conditions, distribution of H3K9cr is almost indistinguishable from that of H3K9ac in HeLa cells (Fig. R5).

Fig. R5. Venn diagram of overlap of H3K9ac and H3K9cr ChIP-seq peaks in HeLa cells.

Figure 5a

How do the authors account for the lack of reduction in H3K14ac after YEATS2 KD?

YEATS2 KD greatly reduces H3K9ac levels, but only mildly affects H3K14ac levels (Fig. 5a). This result indicates that in H1299 cells GCN5/PCAF within the ATAC complex mainly acetylates histone H3K9. However, we cannot exclude the possibility that some other HAT enzymes may compensate the YEATS2 KD defects. For instance, it has been reported that the Hbo1 complex is a major H3K14 HAT in erythroblasts (Mishima et al., 2011). Notably, consistent with our observation in H1299 cells, double KO of GCN5 and PCAF affects mainly the H3K9ac levels, but not the H3K14ac levels in immortalized MEF cells (Jin et al., 2011).

References

- Fields, A.P., Justilien, V., and Murray, N.R. (2016). The chromosome 3q26 OncCassette: A multigenic driver of human cancer. *Adv Biol Regul* 60, 47-63.
- Fong, K.M., and Minna, J.D. (2002). Molecular biology of lung cancer: clinical implications. *Clin Chest Med* 23, 83-101.
- Jin, Q., Yu, L.R., Wang, L., Zhang, Z., Kasper, L.H., Lee, J.E., Wang, C., Brindle, P.K., Dent, S.Y., and Ge, K. (2011). Distinct roles of GCN5/PCAF-mediated H3K9ac and CBP/p300-mediated H3K18/27ac in nuclear receptor transactivation. *The EMBO journal* 30, 249-262.
- Larsen, J.E., and Minna, J.D. (2011). Molecular biology of lung cancer: clinical implications. *Clin Chest Med* 32, 703-740.
- Li, Y., Wen, H., Xi, Y., Tanaka, K., Wang, H., Peng, D., Ren, Y., Jin, Q., Dent, S.Y., Li, W., et al. (2014). AF9 YEATS domain links histone acetylation to DOT1L-mediated H3K79 methylation. *Cell* 159, 558-571.
- Marmorstein, R. (2001). Structure and function of histone acetyltransferases. *Cellular and Molecular Life Sciences* 58, 693-703.
- Mishima, Y., Miyagi, S., Saraya, A., Negishi, M., Endoh, M., Endo, T.A., Toyoda, T., Shinga, J., Katsumoto, T., Chiba, T., et al. (2011). The Hbo1-Brd1/Brpf2 complex is responsible for global acetylation of H3K14 and required for fetal liver erythropoiesis. *Blood* 118, 2443-2453.
- Qian, J., and Massion, P.P. (2008). Role of chromosome 3q amplification in lung cancer. *J Thorac Oncol* 3, 212-215.
- Qian, J., Zou, Y., Wang, J., Zhang, B., and Massion, P.P. (2015). Global gene expression profiling reveals a suppressed immune response pathway associated with 3q amplification in squamous carcinoma of the lung. *Genom Data* 5, 272-274.
- Sequist, L.V., Heist, R.S., Shaw, A.T., Fidias, P., Temel, J.S., Lennes, I.T., Bast, E., Waltman, B.A., Lanuti, M., Muzikansky, A., et al. (2011). SNaPshot genotyping of non-small cell lung cancers (NSCLC) in clinical practice. *J Clin Oncol* 29.
- Szasz, A.M., Lanczky, A., Nagy, A., Forster, S., Hark, K., Green, J.E., Boussioutas, A., Busuttill, R., Szabo, A., and Gyorfy, B. (2016). Cross-validation of survival associated biomarkers in gastric cancer using transcriptomic data of 1,065 patients. *Oncotarget* 7, 49322-49333.
- Umemura, S., Tsuchihara, K., and Goto, K. (2015). Genomic profiling of small-cell lung cancer: the era of targeted therapies. *Jpn J Clin Oncol* 45, 513-519.
- Wan, L., Wen, H., Li, Y., Lyu, J., Xi, Y., Hoshii, T., Joseph, J.K., Wang, X., Loh, Y.E., Erb, M.A., et al. (2017). ENL links histone acetylation to oncogenic gene expression in acute myeloid leukaemia. *Nature* 543, 265-269.

REVIEWERS' COMMENTS:

Reviewer #1 (Remarks to the Author):

The raised concerns have been addressed.

Reviewer #2 (Remarks to the Author):

IN "YEATS2 links histone acetylation to tumorigenesis of non-small cell lung cancer" the authors have address the majority of the initial concerns. The manuscript describes an important role and function for YEATS2 and association with lung cancer as well as other cancers.

In the manuscript, the authors should mention that the G1 arrest could also reflect the impact on DNA replication genes altered by YEATS2 knockdown. They also should mention that the increased HDAC1 is opposite to the increased K9 acetylation. It is not directly mentioned but should be so readers see recognize this fact. Otherwise this manuscript was well executed and nicely controlled.

Reviewer #3 (Remarks to the Author):

The authors have addressed all our concerns in the revised manuscript.

Responses to Reviewers

Reviewer #1 (Remarks to the Author):

The raised concerns have been addressed.

We thank this reviewer for insightful and positive comments.

Reviewer #2 (Remarks to the Author):

IN "YEATS2 links histone acetylation to tumorigenesis of non-small cell lung cancer" the authors have address the majority of the initial concerns. The manuscript describes an important role and function for YEATS2 and association with lung cancer as well as other cancers.

We thank this reviewer for insightful and positive comments.

In the manuscript, the authors should mention that the G1 arrest could also reflect the impact on DNA replication genes altered by YEATS2 knockdown. They also should mention that the increased HDAC1 is opposite to the increased K9 acetylation. It is not directly mentioned but should be so readers see recognize this fact. Otherwise this manuscript was well executed and nicely controlled.

We thank this reviewer for the suggestions. We have included these statements in lines 125-126 and 196.

Reviewer #3 (Remarks to the Author):

The authors have addressed all our concerns in the revised manuscript.

We thank this reviewer for insightful and positive comments.